# Blood meal acquisition enhances arbovirus replication in mosquitoes through activation of the GABAergic system

Yibin Zhu[1,2,3], Rudian Zhang [1,3], Bei Zhang[1], Tongyan Zhao[4], Penghua Wang[5], Guodong Liang[6,7] & Gong Cheng[1,2]

Mosquitoes are hematophagous insects that carry-on and transmit many human viruses. However, little information is available regarding the common mechanisms underlying the infection of mosquitoes by these viruses. In this study, we reveal that the hematophagous nature of mosquitoes contributes to arboviral infection after a blood meal, which suppresses antiviral innate immunity by activating the GABAergic pathway. dsRNA-mediated interruption of the GABA signaling and blockage of the $GABA_A$ receptor by the specific inhibitors both significantly impaired arbovirus replication. Consistently, inoculation of GABA enhanced arboviral infection, indicating that GABA signaling facilitates the arboviral infection of mosquitoes. The ingestion of blood by mosquitoes resulted in robust GABA production from glutamic acid derived from blood protein digestion. The oral introduction of glutamic acid increased virus acquisition by mosquitoes via activation of the GABAergic system. Our study reveals that blood meals enhance arbovirus replication in mosquitoes through activation of the GABAergic system.

[1] Tsinghua-Peking Center for Life Sciences, School of Medicine, Tsinghua University, Beijing 100084, China. [2] Institute of pathogenic organisms, Shenzhen Center for Disease Control and Prevention, Shenzhen, Guangdong 518055, China. [3] School of Life Science, Tsinghua University, Beijing 100084, China. [4] State Key Laboratory of Pathogen and Biosecurity, Beijing Institute of Microbiology and Epidemiology, Beijing 100071, China. [5] Department of Microbiology and Immunology, School of Medicine, New York Medical College, Valhalla, NY 10595, USA. [6] State Key Laboratory of Infectious Disease Prevention and Control, National Institute for Viral Disease Control and Prevention, Chinese Center for Viral Disease Control and Prevention, Beijing 102206, China. [7] Collaborative Innovation Center for Diagnosis and Treatment of Infectious Diseases, Hangzhou 310000, China. Correspondence and requests for materials should be addressed to G.C. (email: gongcheng@mail.tsinghua.edu.cn)

Mosquitoes, belonging to the *Culicidae* family, are a group of hematophagous insects that transmit many human viruses in nature. Feeding on blood is a behavioral trait of mosquitoes that allows them to obtain the nutrients necessary for reproduction[1]. However, mosquitoes may incidentally bite on virus-infected hosts and acquire the viruses circulating in their blood. The viruses subsequently infect and spread systematically in mosquito tissues, such as the salivary glands and neural system. The infected mosquitoes are then ready to transmit the virus to other hosts through blood feeding[2]. Mosquito-borne viruses, which are etiological agents of severe human diseases such as hemorrhagic fever, biphasic fever, encephalitis, and meningitis, cause hundreds of millions of infections and a large number of deaths annually[3–5]. Most mosquito-borne human viruses are categorized into the *Flaviviridae*, *Togaviridae*, and *Bunyaviridae* families[2]. As mosquitoes are primary vectors for the transmission of these viruses, we speculate that the mosquito-borne human viruses may exploit some common mechanisms to facilitate their infections in mosquitoes. However, to date, little information is available regarding these mechanisms.

The GABAergic system is an inhibitory neurotransmitter system that decreases neuronal excitability in insects and mammals[6–8]. Gama-aminobutyric acid (GABA) is generated via decarboxylation of the amino acid glutamic acid by the enzyme glutamic acid decarboxylase (GAD)[9]. Subsequently, GABA is released into the extracellular milieu via exocytosis or reverse transport by GABA transporters. GABA activates the $GABA_A$ receptors, which are ion channels, and the $GABA_B$ receptors, such as G-protein-coupled receptors to conduct its neuro-inhibitory signaling[10]. In addition to its role in neural transmission signaling, the GABAergic system acts as an important player in mammalian immune responses. Many components of the GABAergic system are highly expressed in human lymphocytes to produce and sense GABA, thereby affecting a variety of functional properties of the immune cells such as cytokine secretion, cell proliferation, migration, phagocytic activity, and chemotaxis[11–20]. Furthermore, a recent study demonstrated that the hypermigratory properties of dendritic cells mediated by GABAergic signaling facilitated infection with an intercellular parasite, *Toxoplasma gondii*, in mammals[21]. In insects, the GABAergic system is highly conserved and shows essential activity in neural physiology. Many insecticides, such as Dieldrin[22], Fipronil[23], and Bilobalide[24], have been developed to interfere with insect GABAergic components. Nonetheless, the immune regulatory role of the insect GABAergic system has not been elucidated.

Mosquito-borne viruses are transmitted worldwide and incur an overwhelming public health burden in the tropical and subtropical areas[25]. Understanding of mosquito-virus interactions contributes novel strategies to limit rapid viral transmission in nature and to decrease disease burden. In this study, we found that the GABAergic system could facilitate the infection of many arboviruses by suppressing the innate immune signaling in mosquitoes. The ingestion of blood by mosquitoes efficiently activated the GABAergic system, by enhancing GABA synthesis from glutamic acid as a result of blood digestion, thereby facilitating effective viral infections in mosquitoes. Thus, we demonstrated that the hematophagous nature of mosquitoes contributes to arbovirus infection by activating GABA synthesis.

## Results

### Gene regulation by arbovirus infections in *Aedes aegypti*.
Mosquitoes are the natural vectors for many human viruses in the *Flaviviridae*, *Togaviridae*, and *Bunyaviridae* families[2]. To assess the responses of mosquitoes to infection with various viruses, we selected six mosquito-borne viruses belonging to three virus genera, including dengue virus (DENV) (*Flavivirus*, *Flaviviridae* family), Japanese encephalitis virus (JEV) (*Flavivirus*, *Flaviviridae* family), Sindbis virus (SINV) (*Alphavirus*, *Togaviridae* family), Semliki Forest virus (SFV) (*Alphavirus*, *Togaviridae* family), Batai virus (BATV) (*Orthobunyavirus*, *Bunyaviridae* family), and Tahyna virus (TAHV) (*Orthobunyavirus*, *Bunyaviridae* family), and infected female *A. aegypti* mosquitoes via thoracic microinjection. Mosquitoes inoculated with PBS were used as negative controls. On the basis of the replication rates of these different viruses in mosquitoes (Supplementary Fig. 1), the gene expression on days 1 and 6 post-infection, which represented the early and late time points of infection in the mosquitoes, respectively, was determined by RNA-Seq analyses (Fig. 1a). The number of genes that were up- or down-regulated by each pair of viruses from the same genus are presented in a Venn diagram (Fig. 1b). Compared with the relatively small number of genes with altered expression on day 1 after infection, a larger number of genes were altered on day 6 after infection. Six days post infection in *A. aegypti*, the messenger RNA (mRNA) expression levels of 73, 206, and 62 genes were induced by infection with *Flavivirus*, *Alphavirus*, and *Orthobunyavirus*, and the expression of 11, 16, and 4 genes were impaired by *Flavivirus*, *Alphavirus*, and *Orthobunyavirus*, respectively (Fig. 1b). Interestingly, nineteen up-regulated genes were common to all infection groups, *Flavivirus*, *Alphavirus*, and *Orthobunyavirus* (Table 1). However, none of the down-regulated genes were shared among the three groups on day 6 post infection (Fig. 1b).

### A role for GABAergic system in arboviral infections.
We examined the physiological role of these 19 up-regulated genes in DENV-2 infection. The genes were knocked down individually using double-stranded RNA (dsRNA). DENV-2 was then inoculated into the gene-silenced mosquitoes, and the resultant viral loads were quantified by quantitative polymerase chain reaction (qPCR) 3 days after infection. Knockdown of either *AAEL018153* or *AAEL000405* (Table 1) significantly enhanced the DENV-2 burden compared with green fluorescent protein (GFP) dsRNA treatment (Supplementary Fig. 2). *AAEL018153* encodes a hypothetical protein, and *AAEL000405* encodes a membrane-anchored cell surface protein named *odd Oz*, which is essential for normal retinal and nervous system development[26]. In contrast, silencing of the gene encoding the *A. aegypti* $GABA_A$ receptor component (*AaGABA$_A$-R1*), *AAEL008354*, largely suppressed DENV replication, as measured by qPCR (Supplementary Fig. 2 and Fig. 2a). The viral load in the mosquitoes was validated by a plaque assay (Fig. 2b), suggesting that the *AaGABA$_A$-R1* gene plays a role in the susceptibility of *A. aegypti* to DENV-2 infection. The expression of *AaGABA$_A$-R1* was significantly decreased following dsRNA treatment (Fig. 2c), indicating that the impairment of DENV-2 infection was correlated to the reduction in *AaGABA$_A$-R1* expression. Because previous studies demonstrated that GABAergic signaling plays an effective immune-modulatory role in mammals[13, 15] and contributes to *T. gondii* pathogenesis[21], we focused on *AaGABA$_A$-R1* in the present study.

To determine the pivotal nature of *AaGABA$_A$-R1* to arboviruses of other genera, we microinjected SINV (*Alphavirus*) and TAHV (*Orthobunyavirus*) into *AaGABA$_A$-R1*-silenced mosquitoes and assessed their viral loads 3 days post-infection by qPCR. Both viral loads were impaired by 2–4-fold in *AaGABA$_A$-R1* dsRNA-treated mosquitoes compared with the *GFP*-dsRNA group (Supplementary Fig. 3a and b), implicating a general role of the mosquito GABAergic system in arbovirus replication. However, we have noted that *Culex* mosquitoes are the native vectors for JEV, SINV, and TAHV[27–29]. BATV is transmitted by

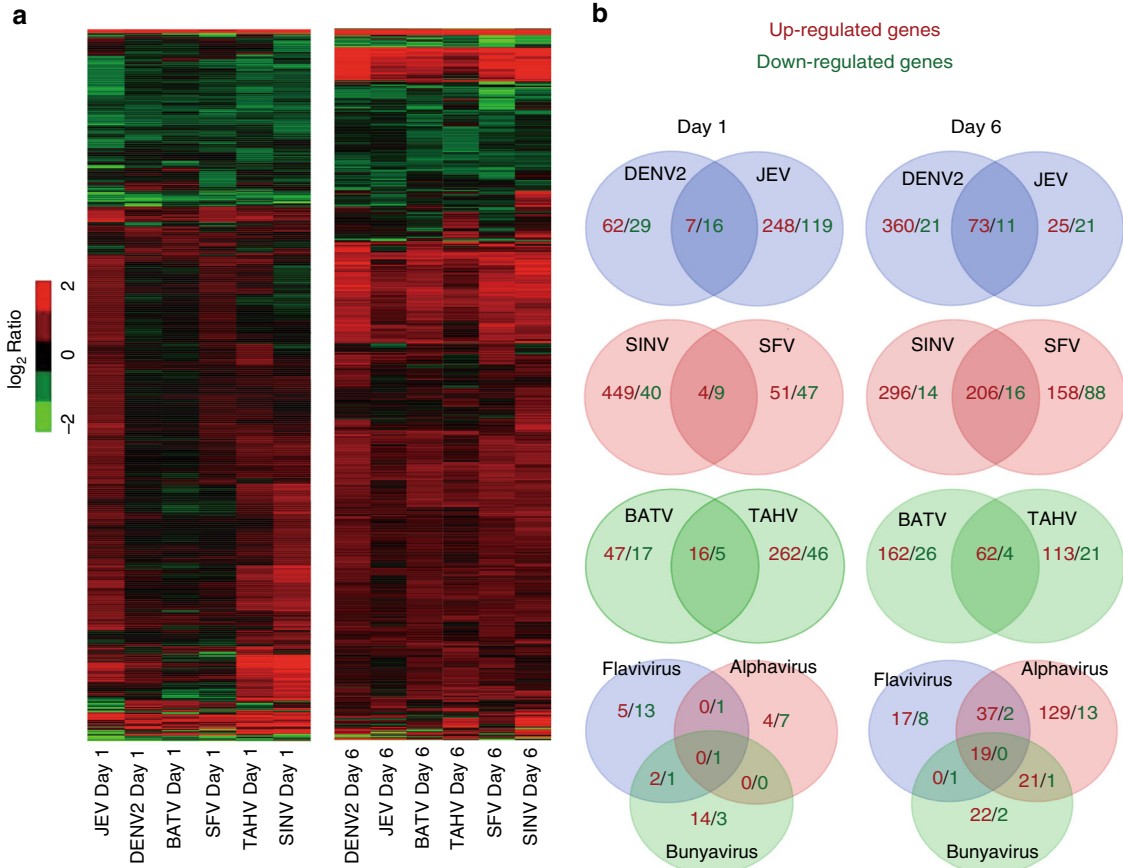

**Fig. 1** Genes regulated by *Flavivirus*, *Alphavirus* a *Orthobunyavirus* infections in *A. aegypti*. **a** Genes regulated by infections with various viruses in *A. aegypti*. For RNA-Seq analysis, total RNA was extracted with TRIzol from the mosquitoes infected with 100 M.I.D.$_{50}$ of viruses on 1 and 6 days post-viral inoculation. The log$_2$ ratio (read number in the virus-induced whole mosquito/read number in the control whole mosquito) was exploited to evaluate gene regulation. Genes with a log$_2$ ratio $\leq-1.5$ or $\geq 1.5$ were selected for further analysis. **b** Venn diagram schematic representation of up-regulated or down-regulated genes by virus infection

| Gene ID | Log2 ratio | | | | | | Gene name |
|---|---|---|---|---|---|---|---|
| | DENV2_D6 vs. Mock | JEV_D6 vs. Mock | SINV_D6 vs. Mock | SFV_D6 vs. Mock | BATV_D6 vs. Mock | TAHV_D6 vs. Mock | |
| AAEL001279 | 9.76 | 11.33 | 10.86 | 10.02 | 10.12 | 8.40 | Merozoite surface protein |
| AAEL007657 | 6.25 | 6.17 | 5.94 | 6.11 | 2.54 | 6.28 | Low-density lipoprotein receptor |
| AAEL004391 | 2.68 | 1.68 | 2.19 | 2.63 | 2.30 | 1.51 | Conserved hypothetical protein |
| AAEL011620 | 2.03 | 2.02 | 2.38 | 1.80 | 1.90 | 2.11 | Conserved hypothetical protein |
| AAEL013532 | 2.48 | 1.74 | 2.08 | 2.62 | 2.91 | 2.46 | Hypothetical protein |
| AAEL005318 | 3.62 | 3.38 | 4.49 | 2.43 | 2.52 | 2.05 | Hypothetical protein |
| AAEL005338 | 3.79 | 3.17 | 4.33 | 2.39 | 2.38 | 2.01 | Novex-3 |
| AAEL018153 | 4.41 | 3.47 | 4.65 | 3.49 | 2.39 | 1.63 | Hypothetical protein |
| AAEL007265 | 3.39 | 2.80 | 3.51 | 3.28 | 2.92 | 2.29 | Hypothetical protein |
| AAEL011569 | 2.92 | 2.48 | 3.43 | 3.44 | 3.28 | 2.29 | Conserved hypothetical protein |
| AAEL002876 | 3.28 | 2.58 | 2.78 | 3.45 | 2.53 | 2.34 | Conserved hypothetical protein |
| AAEL004246 | 2.80 | 2.01 | 2.98 | 2.90 | 2.40 | 1.63 | Still life, sif |
| AAEL009847 | 3.00 | 2.31 | 3.24 | 2.92 | 2.15 | 1.66 | Microtubule-associated protein |
| AAEL008354 | 3.17 | 1.80 | 2.84 | 3.00 | 2.43 | 1.86 | Gaba receptor invertebrate |
| AAEL000405 | 3.40 | 2.25 | 3.23 | 3.26 | 2.66 | 1.80 | Odd Oz protein |
| AAEL005529 | 2.93 | 2.02 | 3.74 | 2.97 | 2.41 | 1.93 | Hypothetical protein |
| AAEL006019 | 3.38 | 2.51 | 3.58 | 2.61 | 2.35 | 1.52 | Voltage-gated sodium channel |
| AAEL003082 | 3.27 | 2.25 | 3.28 | 2.37 | 2.71 | 1.77 | Kinectin, putative |
| AAEL010757 | 2.51 | 1.76 | 3.33 | 2.12 | 2.21 | 1.63 | Hypothetical protein |

**Table 1 Information for the 19 induced genes consistently identified at 6 days post-infection with arboviruses**

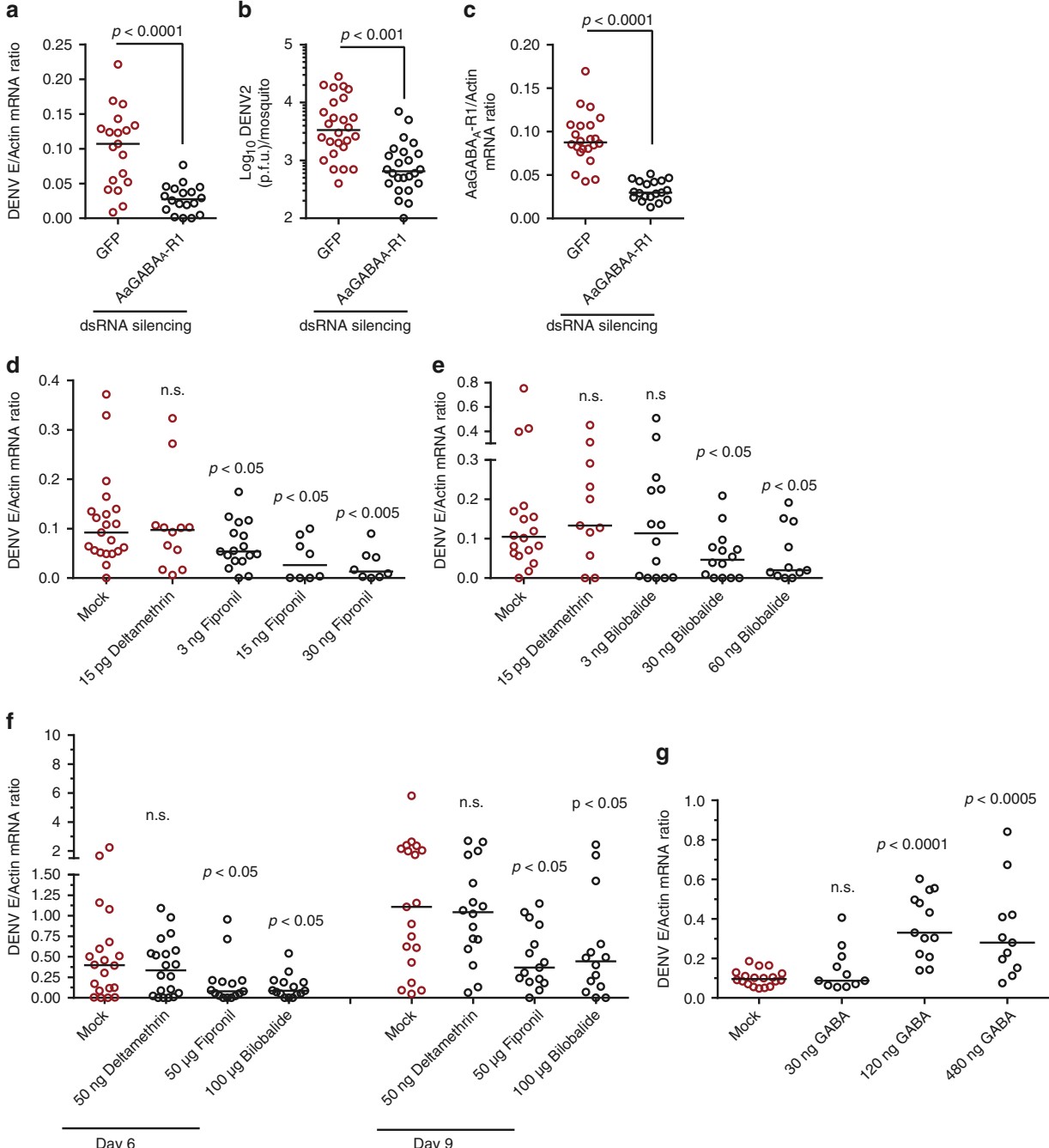

**Fig. 2** Role of GABA receptor-mediated GABA signaling during infection by mosquito-borne viruses. **a–c** dsRNA-mediated knockdown of the *AaGABA_A-R1* gene impaired DENV-2 infection in *A. aegypti*. **a**, **b** Silencing of *AaGABA_A-R1* enhanced DENV-2 infection as measured by qPCR **a** and a plaque assay **b**. The viral loads were assessed on 6 days post-infection via a plaque assay. **c** dsRNA-mediated knockdown of the *AaGABA_A-R1* gene in mosquitoes. Mosquitoes inoculated with *GFP* dsRNA served as negative controls. The *AaGABA_A-R1* abundance was assessed by SYBR Green qPCR on 6 days post dsRNA microinjection. Ten M.I.D._{50} of DENV-2 were inoculated on 3 days post dsRNA inoculation. **d**, **e** Thoracic inoculation of GABA receptor-inhibitory insecticides decreased DENV-2 replication in *A. aegypti*. Either Fipronil **d** or Bilobalide **e** was premixed with 10 M.I.D._{50} of DENV-2 and subsequently inoculated into mosquitoes. Mosquitoes inoculated with Deltamethrin and 10 M.I.D._{50} of DENV-2 served as unrelated controls. Mosquitoes inoculated with PBS and DENV-2 served as mock controls. The number on the X-axis represent the amount of insecticides inoculated per mosquitoes. **f** Insecticide exposure reduced DENV-2 replication in *A. aegypti*. Mosquitoes infected by oral membrane feeding were exposed to bottles sprayed with Fipronil (50 μg/bottle) or Bilobalide (100 μg/bottle) for 2 h. Deltamethrin (50 ng/bottle) served as an unrelated control. Infected mosquitoes exposed to PBS served as mock controls. For mosquito oral infection, $6 \times 10^5$ p.f.u./ml of DENV-2 was used. Surviving mosquitoes were transferred to new culture containers for further rearing. The viral loads were assessed over time post-infection via TaqMan qPCR and were normalized to *A. aegypti* actin (*AAEL011197*). **g** Thoracic inoculation of GABA facilitated DENV-2 infection in *A. aegypti*. Serial concentrations of GABA with 10 M.I.D._{50} of DENV-2 were microinjected into mosquito thoraxes. Mosquitoes inoculated with PBS and DENV-2 served as negative controls. The number on the *X*-axis represents the amount of GABA inoculated per mosquito. **a**, **d**, **e**, **g** The viral loads were assessed at 3 days post-infection via TaqMan qPCR and were normalized to *A. aegypti* actin (*AAEL011197*). **a–g** One dot represents one mosquito, and the horizontal line represents the median of the results. The data were analyzed statistically using the non-parametric Mann–Whitney test. The results were reproduced at least two times. The primers and probes used for PCR are presented in Supplementary Data 2

both *Anopheles*[30] and *Culex*[31] *spp.* in nature. We therefore determined the regulation of the *GABA_A-R1* gene in *Culex pipiens pallens* (*CpGABA_A-R1*, CPIJ008419) infected by JEV, SINV, TAHV, and BATV individually. The *CpGABA_A-R1* gene showed the similar regulation patterns between the infected *C. pipiens pallens* and *A. aegypti*, suggesting that *A. aegypti* is an appropriate mosquito model to assess gene regulation and function with arbovirus infections (Supplementary Fig. 4).

The GABA_A receptors are complex heteropentameric chloride ion channels consisting of several heterotypic subunits[32]. We identified seven homologs of GABA_A receptor components in the *A. aegypti* (*AaGABA_A-R*) proteome (Supplementary Fig. 5a) based on human GABA_A receptor sequences. Next, we determined the role of these *AaGABA_A-R* genes in both DENV-2 (Supplementary Fig. 5b) and SINV (Supplementary Fig. 5c) infections via thoracic dsRNA microinjection. In accordance with *AaGABA_A-R1*, dsRNA-mediated knockdown of the other 6 *AaGABA_A-R* genes consistently reduced both viral burdens in *A. aegypti*, demonstrating that GABA signaling may facilitate arbovirus infection in mosquitoes.

Next, we used GABA receptor inhibitors to validate the afore-mentioned gene silencing results. Several commercial insecticides, such as Fipronil[23] and Bilobalide[24], are inhibitors of insect GABA_A receptors and disrupt GABA signaling. We selected Deltamethrin as a control insecticide that does not target on the GABAergic system[33]. Thoracic inoculation of Fipronil (Supplementary Fig. 6a), Bilobalide (Supplementary Fig. 6b), or Deltamethrin (Supplementary Fig. 6c) all resulted in high *A. aegypti* mortality in an insecticide-dose dependent manner, although some mosquitoes were refractory to the insecticides and survived. We next thoracically co-microinjected the insecticides and DENV-2 into mosquitoes. Compared with that of the control mosquitoes, there was no change in DENV-2 burden in the Deltamethrin-treated surviving *A. aegypti* (Fig. 2d and e). However, inoculation of Fipronil (Fig. 2d) and Bilobalide (Fig. 2e) significantly reduced viral replication in the surviving mosquitoes. Both of these GABA_A-receptor targeting insecticides did not have additional effects on DENV-2 replication in the *AaGABA_A-R1*-silenced mosquitoes (Supplementary Fig. 7a and b), suggesting that the insecticides impair the viral replication by inhibiting GABA signaling. To validate the phenotypes in relatively natural settings, the *A. aegypti* mosquitoes that had been infected with DENV-2 orally via a blood meal were maintained in the containers sprayed with either Fipronil (50 μg/bottle), Bilobalide (100 μg/bottle) or Deltamethrin (50 ng/bottle), respectively. Exposure to either Fipronil (Fig. 2f) or Bilobalide (Fig. 2f), but not Deltamethrin, reduced the DENV burden in the surviving mosquitoes over the time course of insecticide treatment, further validating that the blockage of mosquito GABA signaling by these insecticides reduced arbovirus infection.

Recent outbreaks of Zika virus (ZIKV), a mosquito-borne flavivirus, have introduced an overwhelming burden to global public health. Therefore, we assessed the role of GABA signaling in ZIKV mosquito infection by microinjection. Expression of the *AaGABA_A-R1* gene significantly increased in response to ZIKV infection (Supplementary Fig. 8a). Either dsRNA-mediated interference of *AaGABA_A-R1* (Supplementary Fig. 8b) expression or blockage of *AaGABA_A-R* signaling by Fipronil (Supplementary Fig. 8c) and Bilobalide (Supplementary Fig. 8c) consistently reduced ZIKV replication in *A. aegypti* infected by microinjection or blood feeding. Taken together, these results clearly demonstrate an important role for GABA signaling in the common arboviral infections of *A. aegypti*.

GABA is a chemical transmitter that is recognized by GABA receptors to trigger intracellular signaling[10]. To further validate the role of the GABA signaling pathway in arbovirus infection, we inoculated serial concentrations of GABA together with DENV-2 into *A. aegypti*. Thoracic microinjection of GABA did not largely influence the survival rates of mosquitoes (Supplementary Fig. 9). However, the GABA inoculation significantly enhanced the DENV-2 burden compared with the mock group (Fig. 2g), and also enhanced SINV (Supplementary Fig. 10a) and TAHV (Supplementary Fig. 10b) infections as well, further validating the role of mosquito GABA signaling in arbovirus infections.

GABA is produced from glutamic acid by glutamic acid decarboxylases (GADs) in the cytoplasm[34] and is then released into the extracellular milieu via exocytosis or reverse-transported by GABA transporters (Fig. 3a). We identified two *GAD* homologs in the *A. aegypti* genome, designated *A. aegypti GAD 1* (AAEL011981, *AaGAD1*) and *GAD 2* (AAEL007542, *AaGAD2*). Because the transcript level of *AaGAD2* was too low to be measured by qPCR, we speculated that *AaGAD2* might not be essential for GABA production and therefore focused on *AaGAD1* for our investigation. Intriguingly, expression of the *AaGAD1* gene was induced by DENV-2 infection (Fig. 3b). Along with the up-regulation of the *AaGAD1* gene, GABA production also increased over the time course of infection (Fig. 3c). Consistently, dsRNA-mediated knockdown of the *AaGAD1* gene impaired the replication of DENV-2 (Fig. 3d), SINV (Fig. 3e), and TAHV (Fig. 3f), in *A. aegypti*. In the *AaGAD1*-silenced mosquitoes, supplementation of GABA via thoracic microinjection restored viral infection to the level of *GFP* dsRNA-treated *A. aegypti* (Fig. 3d and f). Altogether these data convincingly demonstrate that the GABAergic system commonly facilitates the infection of human arboviruses in mosquitoes.

**GABAergic system suppresses the Imd pathway in mosquito.** To investigate the molecular mechanism by which the GABAergic system facilitates arbovirus infection, we exploited two independent dsRNAs to knockdown the *AaGABA_A-R1* gene in *A. aegypti* (Supplementary Fig. 11a and b). *GFP* dsRNA-treated mosquitoes served as controls. RNA-Seq and in-depth analysis of immune-related genes revealed that the up-regulated genes and down-regulated genes were consistently regulated by these independent dsRNA treatments (Fig. 4a and Supplementary Data 1). Intriguingly, many genes encoding components and effectors of the Imd (Immune Deficiency) pathway, such as *A. aegypti Imd* (*AaImd*) (Fig. 4b), *Rel2* (*AaRel2*) (Fig. 4c), *Defensins* (*AaDef*) (Fig. 4d–f), and *Cecropin* N (*AaCec-N*) (Fig. 4g), were significantly induced in the *AaGABA_A-R1*-silenced mosquitoes. However, the component genes of the RNA interference (RNAi) (*Dicer2* and *Argonaute2*)[35, 36], Toll (*Toll* and *Myd88*)[37], and JAK-STAT pathway (*STAT1* and *Dome*)[2] pathways were not up-regulated by both *AaGABA_A-R1* silencing (Supplementary Data 1). We next assessed whether feeding with GABA may affects the Imd pathway. As proliferation of the microbiota can skew the Imd signaling, we removed the gut commensal microbiome by feeding antibiotics[38]. A sucrose meal with 120 ng of GABA reduced the expression of the *AaImd*, *AaRel2*, *AaDef-C*, and *AaCec-N* genes in the midgut of antibiotic-treated mosquitoes (Supplementary Fig. 12a–d). Consistently, a sucrose meal with 120 ng of GABA enhanced the burden of commensal bacteria in the mosquito guts (Supplementary Fig. 12e), indicating the GABA signaling negatively regulates the Imd-mediated gut immunity.

To further assess the role of the Imd pathway in arbovirus infection, we silenced the *AaImd* gene via thoracic dsRNA microinjection (Fig. 4h) and subsequently inoculated DENV-2 (Fig. 4i), SINV (Fig. 4j), or TAHV (Fig. 4k) into the gene-silenced mosquitoes. The resultant viral loads were quantified by qPCR 3 days after infection. In accordance with the results obtained for the *AaGABA_A-R1*-silenced mosquitoes, genetic suppression of the

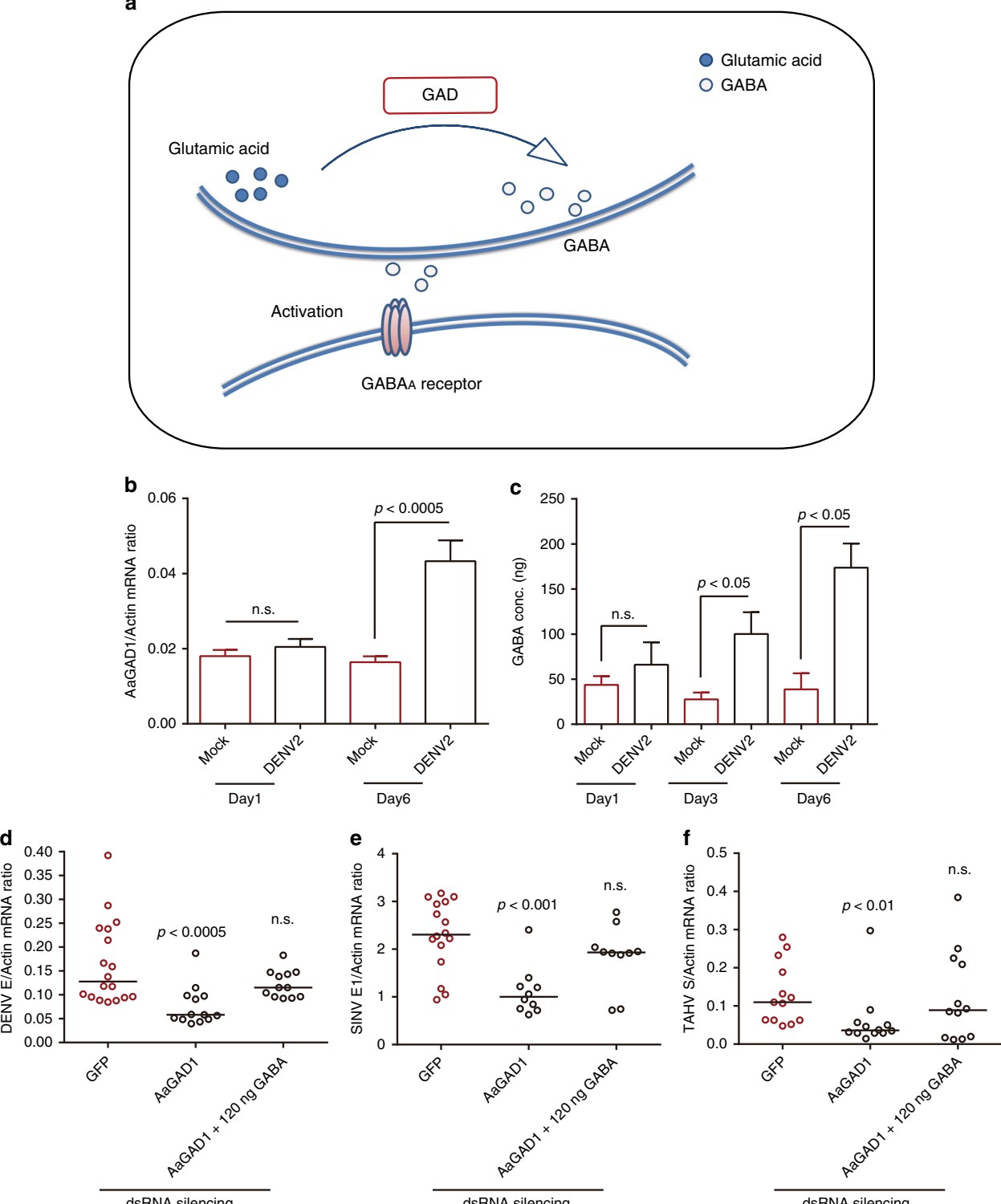

**Fig. 3** Role of the GABA production system during infection with mosquito-borne viruses. **a** Schematic representation of the GABAergic system. **b**, **c** GABA generation was induced by DENV-2 infection in *A. aegypti*. **b** The expression of *AaGAD1* was enhanced by DENV-2 infection. The abundance of *AaGAD1* was assessed by SYBR Green qPCR on 1 and 6 days post-infection. **c** GABA induction by DENV-2 infection in *A. aegypti*. Ten M.I.D.$_{50}$ of DENV-2 were inoculated into the thoracic region of mosquitoes. Mosquitoes that were microinjected with PBS served as mock controls. The data are presented as the mean ± s.e.m. The results were reproduced three times. **d–f** Role of the GABA production system during infection with mosquito-borne viruses. *AaGAD1* was silenced by inoculation with *AaGAD1* dsRNA. Mosquitoes inoculated with *GFP* dsRNA served as negative controls. Three days post dsRNA treatment, 10 M.I.D.$_{50}$ of DENV-2 **d**, SINV **e** or TAHV **f**, with or without 120 ng of GABA, were microinjected into *A. aegypti* mosquitoes. The viral loads were assessed at 3 days post-infection via TaqMan qPCR **d** or SYBR Green qPCR **e**, **f** and were normalized to *A. aegypti* actin (*AAEL011197*). The primers and probes used for PCR are presented in Supplementary Data 2. One dot represents 1 mosquito, and the horizontal line represents the median of the results. **d–f** The number on the X-axis represents the amount of GABA inoculated per mosquito. **b–f** The data were analyzed statistically using the non-parametric Mann–Whitney test. The results were reproduced three times

Imd pathway significantly enhanced viral replication in *A. aegypti* (Fig. 4i–k), indicating the role of the Imd pathway in the resistance to arboviral infections in mosquitoes. Furthermore, activation of Imd signaling by peptidoglycan (PGN)[39] inoculation rescued viral replication enhancement in the GABA-inoculated mosquitoes (Fig. 4l). The interruption of the Imd signaling offset the reduction of DENV infection in the *AaGABA_A-R1*-silenced mosquitoes (Fig. 4m). Taken together, the GABAergic system

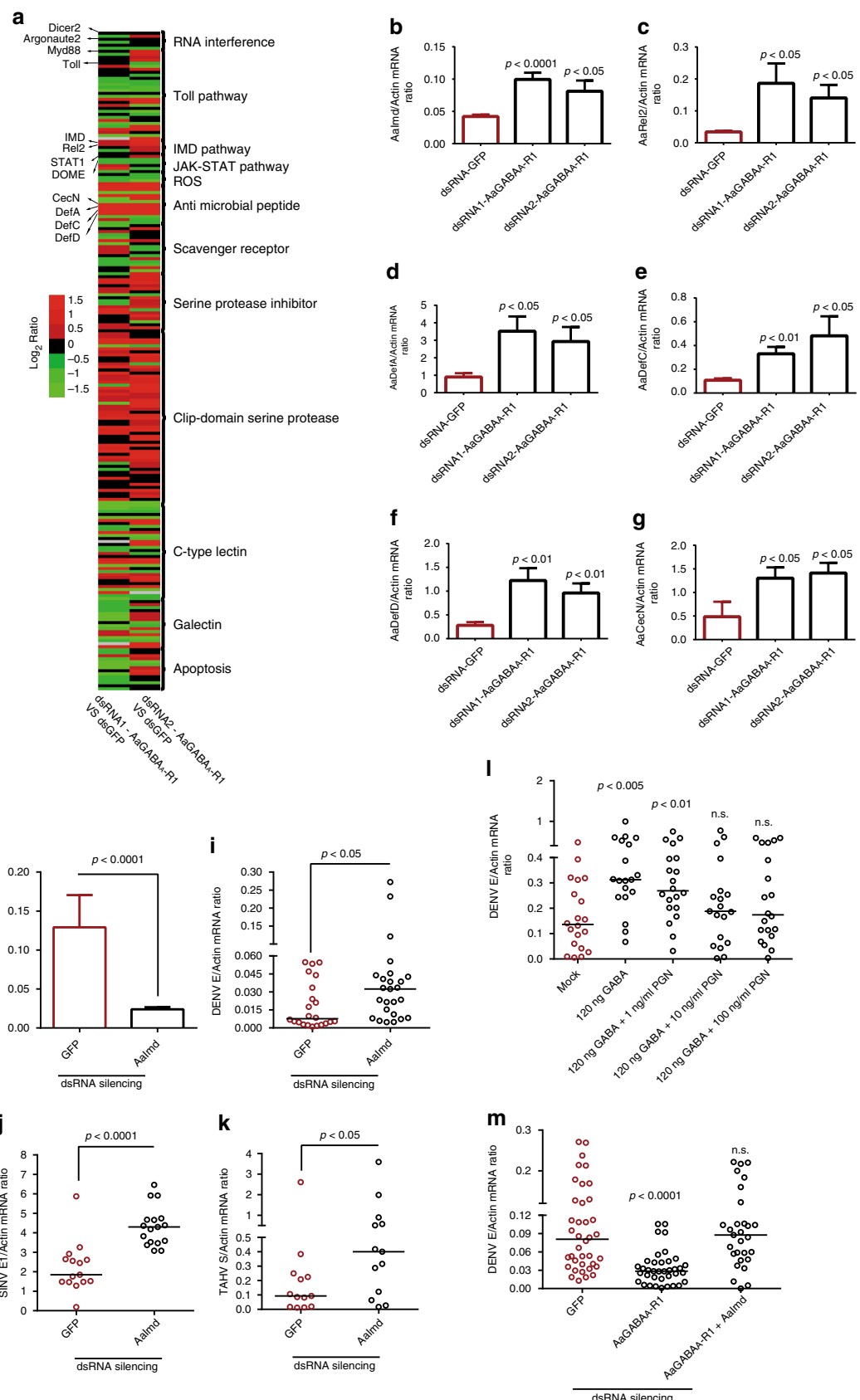

may facilitate arbovirus infection in mosquitoes by suppressing the Imd pathway.

**Blood meal enhances viral infection by GABAergic activation.**
Blood meals are the primary means by which mosquitoes acquire arboviruses from infected hosts. The blood components obtained from hosts are digested into amino acids in the gut lumen and are subsequently absorbed as nutrients by mosquitoes[1]. Therefore, blood feeding may result in the enhancement of GABA signaling through the production of large amounts of glutamic acid from blood digestion. Moreover, during the period of blood digestion, the acquired viruses infect the gut epithelial cells and may subsequently spread systemically into the mosquito hemocoel[2]. Therefore, we speculate that blood feeding may contribute to the infection and dissemination of arboviruses in mosquitoes by activating the GABAergic system. Compared with that of sucrose meals (Supplementary Figs 13a, b and 14)[40–43], blood meals largely enhanced the infection of DENV-2 in *A. aegypti* (Fig. 5a), JEV in *C. pipiens pallens* (Fig. 5b), and TAHV in *C. pipiens pallens* (Fig. 5c). *C. pipiens pallens* is an established mosquito model for the infection study of JEV[44] and TAHV[29]. Silencing *AaGAD1* partially offset the enhancement of DENV infection mediated by blood meals (Supplementary Fig. 15), suggesting that the GABAergic system contributes to the arbovirus infection through blood meals. Next, we evaluated the regulation of GABA signaling in response to the mosquito blood meal. Either whole human blood or the 70 mg/ml BSA, which is equivalent to the total amount of protein in human serum[45], was used to feed *A. aegypti* mosquitoes. Mosquitoes that were fed sucrose served as negative controls. Compared with sucrose meals, the ingestion of blood significantly augmented the amount of glutamic acid (Fig. 5d) and GABA (Fig. 5e) in mosquitoes on days 3 and 5 postmeal. Intriguingly, the ingestion of BSA led to a similar induction (Fig. 5d, e), indicating that the protein digestion from oral feeding contributes to the activation of the GABAergic system in mosquitoes.

We next assessed whether the oral introduction of glutamic acid could regulate the Imd pathway through the GABAergic system. The oral introduction of glutamic acid impaired the expression of the *AaImd*, *AaRel2*, *AaDef-C*, and *AaCec-N* genes in the midgut of antibiotic-treated mosquitoes (Supplementary Fig. 16a–d). However, silencing the *AaGAD1* gene offset these expression impairments (Supplementary Fig. 16a–d), indicating that glutamic acid impairs the Imd pathway through the GABA derived from it and not by directly regulating Imd signaling activity. Subsequently, we assessed whether feeding glutamic acid may affect the microbial proliferation in the mosquito guts. After

a sucrose meal with glutamic acid, the burden of commensal bacteria was enhanced in the mosquito gut. However, knockdown of the *AaGAD1* gene rescued the microbial proliferation caused by glutamic acid introduction (Supplementary Fig. 16e), further indicating that glutamic acid modulates the gut immune activity through the GABAergic system.

We next assessed the effect of oral introduction of glutamic acid in DENV-2 infection in *A. aegypti*. A mixture, which contained 1% sucrose (50% v/v), the supernatant from DENV-2-infected Vero cells (cultured in serum-free medium) (50% v/v), and 10 μg/ml or 100 μg/ml glutamic acid, was used to feed *A. aegypti* via an in vitro blood feeding system. Mosquitoes feeding on the mixture without glutamic acid served as negative controls. The ratios of DENV-2 infection in the mosquitoes were significantly enhanced with the oral introduction of glutamic acid (Fig. 6a). Glutamic acid did not directly affect the infectivity of DENV-2 virions (Supplementary Fig. 17). The phenotype of the DENV-2 viral loads in the mosquitoes was further reproduced by a plaque assay (Supplementary Fig. 18a). Furthermore, the oral introduction of glutamic acid enhanced the ratios of midgut infection (Supplementary Fig. 19a), head dissemination (Supplementary Fig. 19d), and salivary glands infection (Supplementary Fig. 19g), indicating that treatment with glutamic acid may promote the DENV-2 infection and dissemination. Knockdown of the *AaGAD1* gene suppressed glutamic acid-mediated GABA generation (Fig. 6b), and reduced the DENV-2 infective ratios (Fig. 6c) in the glutamic acid-fed *A. aegypti* mosquitoes, further validating that glutamic acid-mediated GABA generation plays an important role in DENV acquisition via mosquito blood meals.

The afore-mentioned studies were all performed with the *A. aegypti* long-term lab-adapted Rockefeller strain and DENV-2 NGC lab-adapted strain. To further validate the role of the GABAergic system in DENV infection via relatively natural settings, we exploited a field strain of *A. aegypti* from the Yunnan province in China[46] and the low-passage clinical DENV-2 43 strain[44] in our investigation. Consistently, the knockdown of *AaGABA_A-R1* impaired DENV-2 43 infection (Supplementary Fig. 20a); however, thoracic inoculation of GABA enhanced the infection (Supplementary Fig. 20b), in the field *A. aegypti* mosquitoes. Compared to that of the sucrose meals, feeding either blood (Supplementary Fig. 20c) or glutamic acid (Supplementary Fig. 20d) largely enhanced DENV-2 43 infection in the field *A. aegypti*, suggesting the GABAergic system may play a general role in the promotion of arbovirus infection in mosquitoes under the natural conditions.

We next extended our observation to infection by other arboviruses in their mosquito vectors. The oral introduction of

**Fig. 4** The GABAergic system suppresses the gene expression of the Imd pathway. **a** Regulation of immune-related genes in the *AaGABA_A-R1*-silenced mosquitoes. Two independent dsRNAs were exploited to knockdown the *AaGABA_A-R1* gene in *A. aegypti*, respectively. The *GFP* dsRNA-treated mosquitoes served as controls. The total RNA was isolated on 3 days post dsRNA inoculation for RNA-Seq analysis. Immune-related genes were clustered according to immune pathways and factors. **b–g** Regulation of *Imd*-related genes by *AaGABA_A-R1* silencing in *A. aegypti*. *AaGABA_A-R1* was silenced by two independent dsRNAs. *GFP* dsRNA-treated mosquitoes served as controls. Expression of *AaImd* **b**, *AaRel2* **c**, *AaDef-A* **d**, *AaDef-C* **e**, *AaDef-D* **f** and *AaCec-N* **g** were determined by SYBR Green qPCR at 3 days post dsRNA microinjection. **h–k** Knockdown of *AaImd* enhanced virus infection in *A. aegypti*. **h** Inoculation of *AaImd* dsRNA significantly suppressed the *AaImd* expression. Mosquitoes inoculated with *GFP* dsRNA served as negative controls. *AaImd* abundance was assessed by SYBR Green qPCR at 6 days post dsRNA microinjection. Silencing *AaImd* enhanced infection of DENV-2 **i**, SINV **j** and TAHV **k** in *A. aegypti*. In total 10 M.I.D._{50} of each virus were inoculated at 3 days post dsRNA inoculation. **l** PGN-mediated activation of Imd signaling offset the increase in DENV replication in GABA-inoculated mosquitoes. Serial concentrations of PGN with 10 M.I.D._{50} of DENV-2, with or without 120 ng of GABA, were co-inoculated into mosquitoes. Mosquitoes inoculated with PBS served as mock controls. **m** Silencing the *Imd* gene offset the reduction in DENV-2 infection in the *AaGABA_A-R1*-silenced mosquitoes. Mosquitoes inoculated with *GFP* dsRNA served as negative controls. Ten M.I.D._{50} of each DENV-2 were inoculated at 3 days post dsRNA inoculation. **b–h** The data are presented as the mean ± s.e.m. The results were reproduced twice. **i–m** The viral loads were assessed at 3 days post-infection via TaqMan qPCR. One dot represents one mosquito and the horizontal line represents the median of the results. **b–m** The genes were normalized to *A. aegypti* actin (*AAEL011197*). The data were analyzed statistically using the non-parametric Mann–Whitney test. The data from at least two independent experiments were combined

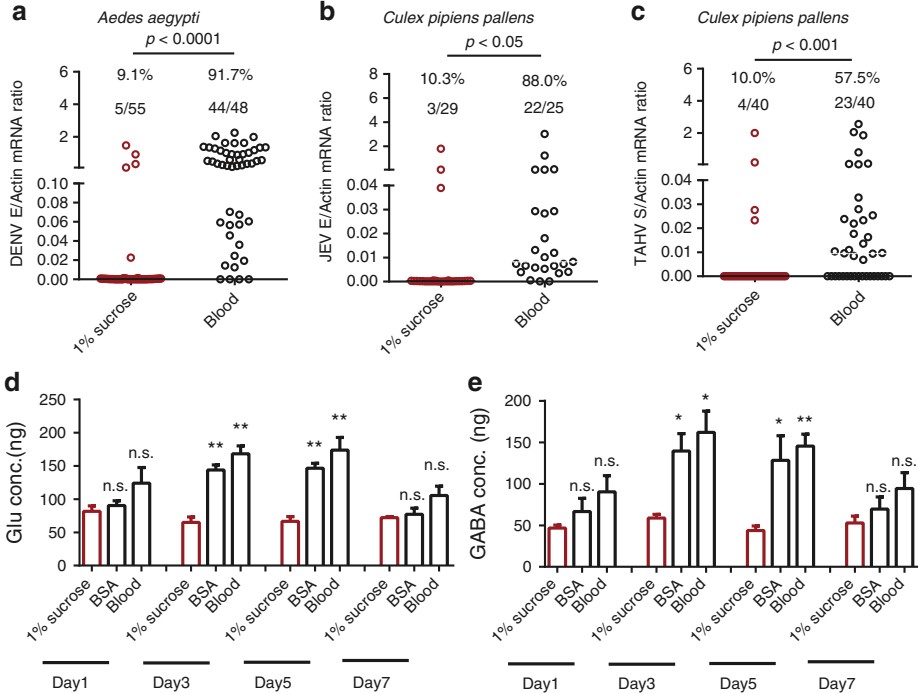

**Fig. 5** Glutamate-mediated GABA induction by blood ingestion. **a–c** Feeding blood enhanced the infection of DENV-2 in *A. aegypti* **a**, JEV in *C. pipiens pallens* **b** and TAHV in *C. pipiens pallens* **c**. Either human blood or 1% sucrose (500 μl) was premixed with supernatant from $6 \times 10^5$ p.f.u./ml of DENV-2-infected **a**, $5 \times 10^5$ p.f.u./ml of JEV-infected **b** or $5 \times 10^6$ p.f.u./ml of TAHV-infected **c** Vero cells (cultured in serum-free medium) (500 μl) to feed mosquitoes via an in vitro membrane blood meal. Mosquito infectivity was determined by TaqMan qPCR at 8 days post-blood meal. The number of infected mosquitoes relative to the total number of mosquitoes is shown at the top of each column. Each dot represents one mosquito. The data upper mosquito number are represented as the percentage of mosquito infection. Differences in the mosquito infective ratios were compared using Fisher's exact test. **d** and **e** Protein ingestion augmented the amount of glutamic acid **d** and GABA **e** in the mosquitoes over time after a meal. Either whole human blood or 70 mg/ml BSA was used for oral feeding of *A. aegypti*. Mosquitoes fed 1% sucrose served as negative controls. The fed mosquitoes were collected over time after feeding and were used to assess the amounts of glutamic acid and GABA present. The numbers on the *Y*-axis represents the amount of GABA inoculated per mosquito. The data are presented as the mean ± S.E.M. Differences were considered significant if $P < 0.05$. *$P < 0.05$; **$P < 0.01$; and ***$P < 0.001$. The data were analyzed statistically using the non-parametric Mann–Whitney test. The results were reproduced twice. **a–e** The data from at least two independent experiments were combined

glutamic acid with a sucrose meal enhanced SINV infection in *A. aegypti*, as measured by qPCR (Fig. 6d) and a plaque assay (Supplementary Fig. 18b). Furthermore, we applied similar procedures to those described above to infect *C. pipiens pallens* with either JEV or TAHV. Consistently, feeding with glutamic acid enhanced the infectious ratios of both JEV (Fig. 6e and Supplementary Fig. 18c) and TAHV (Fig. 6f and Supplementary Fig. 18d) in the fed *Culex* mosquitoes. We next assessed the vector competence in the SINV infection of *A. aegypti* and the TAHV infection of *C. pipiens pallens*. In the mosquitoes feeding with SINV (Supplementary Fig. 19b, e and h) and TAHV (Supplementary Fig. 19c, f and i), the oral introduction of glutamic acid significantly enhanced the ratios of midgut infection and head dissemination, indicating that mosquito hematophagy may enhance GABAergic signaling to commonly facilitate arbovirus infection.

## Discussion

Blood acquisition by feeding on a host is a defining characteristic of hematophagous arthropods. Several hematophagous arthropods, such as mosquitoes, sandflies, ticks, and lice, are common vectors for numerous human pathogens[47]. Through feeding on an infected host, vectors can acquire pathogens together with the host blood. Subsequently, the pathogens infect and spread systematically in the vectors and then replicate in the salivary glands. The infected mosquitoes are then ready to transmit pathogens back to other hosts through bites[2, 48]. Therefore, pathogen

acquisition by arthropod vectors through feeding on an infected host is an essential step in the life cycle of vector-borne pathogens. Given the native relationship between pathogens and their vectors, pathogens may exploit some mechanisms in arthropods to facilitate their infections. For example, mosquito soluble C-type lectins (*mosGCTLs*) recognize flavivirus envelope proteins to facilitate infection in both *A. aegypti* and *Culex quinquefasciatus*[49, 50]. A mosquito protein, CRVP379, is highly induced by DENV infection and might act as a putative DENV receptor[51]. In addition, the pathogens must overcome multiple immune barriers associated with the midgut and hemocoel of the vectors[2, 44, 52, 53]. These immune barriers can interrupt invasion of the pathogens in midgut epithelial cells and modulate their replication to escape into the hemocoel[54, 55]. In this study, we reveal that the hematophagous property of mosquitoes contributes to the replication of arboviruses after a blood meal. Mosquito blood meal results in robust activation of the GABAergic system through glutamate-derived GABA production from blood digestion. The enhancement of GABA signaling suppresses antiviral responses, such as *AMP* induction by the Imd signaling pathway, thereby facilitating viral infection of the mosquito vectors. Therefore, we propose a common mechanism to facilitate the infection of many arboviruses in mosquitoes: a GABAergic signaling-based immune suppression induced by blood ingestion.

The GABAergic system is a major neural inhibitory mechanism. Activation of GABA signaling generally leads to the

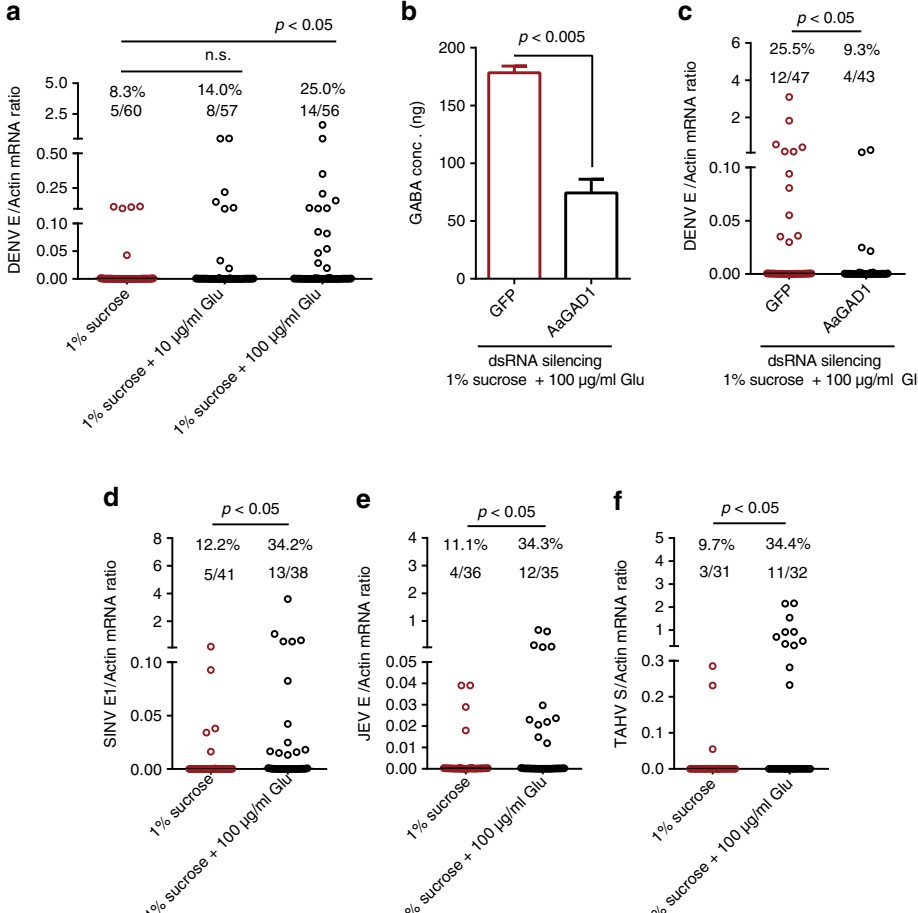

**Fig. 6** Glutamate-mediated GABA generation facilitates infection of mosquito vectors by mosquito-borne viruses. **a** The oral introduction of glutamic acid enhanced the prevalence of DENV-2 infection in *A. aegypti*. A mixture, which contained 1% sucrose (500 μl), supernatant from 6 × 10⁵ p.f.u./ml of DENV-2-infected Vero cells (cultured in serum-free medium) (500 μl), and 10 μg/ml or 100 μg/ml glutamic acid, was used to feed *A. aegypti* via an in vitro blood feeding system. Mosquitoes fed this mixture without glutamic acid served as a negative control. **b, c** Knockdown of the *AaGAD1* gene suppressed glutamate-mediated GABA generation **b**, resulting in reduced DENV infection **c** with sucrose feeding. *AaGAD1* was silenced by inoculation with *AaGAD1* dsRNA. Mosquitoes inoculated with *GFP* dsRNA served as negative controls. Three days post dsRNA treatment, a mixture that containing 1% sucrose (500 μl), supernatant from DENV-2-infected Vero cells (cultured in serum-free medium) (500 μl), and 100 μg/ml glutamic acid, was used to feed *A. aegypti* via an in vitro blood feeding system. **d–f** Sucrose feeding with glutamic acid enhanced the prevalence of SINV infection in *A. aegypti* **d**, JEV infection in *C. pipiens pallens* **e** and TAHV infection in *C. pipiens pallens* **f**. A mixture, which contained 1% sucrose (500 μl), supernatant from either 2 × 10⁶ p.f.u. ml⁻¹ of SINV-infected **d**, 5 × 10⁵ p.f.u./ml of JEV-infected **e** and 5 × 10⁶ p.f.u./ml of TAHV-infected **f** Vero cells (cultured in serum-free medium) (500 μl), and 100 μg/ml glutamic acid, was used to feed mosquitoes via an in vitro blood feeding system. Mosquitoes fed on this mixture without glutamic acid served as negative controls. **b** The numbers on the *Y*-axis represents the amount of GABA inoculated per mosquito. The results were reproduced twice. **a, c–f** Mosquito infectivity was determined by TaqMan qPCR or SYBR Green at 8 days post-blood meal. The number of infected mosquitoes relative to the total numbers of mosquitoes is shown at the top of each column. Each dot represents one mosquito. The data upper mosquito number are represented as the percentage of mosquito infection. Differences in the mosquito infective ratios were compared using Fisher's exact test. **a–f** The data from at least two independent experiments were combined

suppression of neuronal excitability in mature neurons[10]. However, it recently became clear that the GABAergic system plays a vital role in immune regulation. Multiple key GABAergic components have been identified in mammalian T cells, macrophages and dendritic cells[11, 12, 16, 56]. GABA has multiple functions in immune cells, such as the regulation of cytokine secretion[15], modification of proliferation[12] and migration of immune cells[20]. In accordance with its role in the immune system in mammals, we herein revealed that activation of the GABAergic system suppresses the antiviral immune responses in *A. aegypti*. Mosquitoes have evolved efficient antiviral strategies to restrict viral replication. Multiple studies have reported that RNA interference (RNAi) and several other conserved innate immune responses such as the Toll, Imd, JAK-STAT, and complement-like pathways play systemic roles against arbovirus infection in mosquitoes[2].

However, the knockdown of *AaGABA_A-R1* did not affect the expression of molecules in the RNAi and other pathways but instead activated the expression of many Imd components, suggesting the GABAergic system plays a role in arbovirus infection by suppressing the Imd pathway.

The mosquito Imd pathway has been shown to mediate NF-kB-dependent immune responses during the elimination of invading gram-negative bacteria[2, 57, 58]. In the Imd pathway, the cell surface pattern recognition receptor binds microbes and then recruits the adapter molecule Imd, which results in the activation of an NF-kB-like factor denoted Rel2. The activated Rel2 is translocated to the nucleus to induce the transcription of immune effector genes such as antimicrobial peptides (*AMPs*)[58]. In addition to its bactericidal activity, the insect Imd pathway plays important antiviral roles against viral infection in insects[2]. The

Imd pathway can be activated by SINV infection[59] and mediate an antiviral response to SINV replication in *Drosophila* and mosquito cells[60]. DENV infection induces multiple immune components of the Imd pathway in *A. aegypti* salivary glands[60]. In addition, *Anopheles gambiae* infection with o'nyong-nyong virus is enhanced by the knockdown of Imd components in the mosquito midgut[61]. AMPs are a group of immune effectors that are activated by the Imd signaling pathway. Indeed, it is well known that mammalian AMPs efficiently eliminate bacteria, enveloped viruses, and fungi[62]. Given their conserved sequences, insect AMPs also play a crucial role in antiviral defense. A Cec-like peptide was recently found to be induced by DENV infection, limits the virus in the salivary gland of *A. aegypti*[60]. Several AMPs can be induced by DENV infection, and moreover, knockdown of these *AMP* genes results in enhanced DENV replication in *A. aegypti*[63]. In the present study, we found that the expression of several Imd-related *Defensins* and *Cecropins*, which have antiviral activity in mosquitoes[63], was significantly enhanced in *AaGABA*$_A$*-R1*-silenced mosquitoes by independent dsRNAs, suggesting that activation of the GABAergic system may suppress the induction of these antiviral *AMP* genes.

Vector control, such as insecticide spraying, is an important global strategy for eliminating mosquito-borne diseases. Fipronil is an insecticide that targets GABA receptors to interrupt neural transmission, thereby resulting in insect death[6, 7, 64]. In addition, a Ginkgo biloba extract named Bilobalide, which is similar to the GABA$_A$ receptor antagonist picrotoxin, shows insecticidal activity by interacting with GABA receptors[24]. Indeed, insecticide sprays cannot thoroughly eradicate all mosquitoes in practical situations, and the mosquitoes that survive can still carry and transmit viruses to humans. Therefore, limiting of viral replication in the remaining mosquitoes is useful for mosquito-borne disease control. Intriguingly, the application of two insecticides, Fipronil and Bilobalide, both of which act as GABA inhibitors, significantly reduced the viral loads of DENV-2 and ZIKV in the remaining insecticide-treated *A. aegypti* mosquitoes that survive, suggesting that GABA receptor-targeting insecticides may more efficiently control the dissemination of arboviruses by mosquitoes in nature.

Mosquito-borne viral diseases are a major concern for global health and result in significant economic losses in many countries. Understanding the sophisticated interactions between mosquito-borne viruses and their vectors may provide new strategies for disease control. Here, we showed that the GABAergic system facilitates a wide spectrum of arbovirus infections in mosquitoes by suppressing mosquito innate immunity. Blood digestion-mediated GABAergic activation contributes to the replication and dissemination of the acquired viruses in mosquitoes through blood meals. Our study offers a common mechanism that renders mosquitoes permissive to arboviruses through feeding blood. Furthermore, manipulation of the GABAergic system may be a feasible approach to interrupt the arbovirus lifecycle to control dissemination of these viruses in nature.

## Methods

**Ethics statement**. Human blood for mosquito feeding was taken from healthy donors who provided written informed consent. The collection of human blood samples was approved by the local ethics committee at Tsinghua University.

**Mosquitoes, cells and viruses**. *Aedes aegypti* (Rockefeller strain), *Aedes aegypti* (Yunnan strain), and *Culex pipiens pallens* (Beijing strain) mosquitoes, provided by Beijing Institute of Microbiology and Epidemiology, were maintained in a sugar solution at 27 °C and 80% humidity according to standard rearing procedures[38, 44]. DENV-2 (New Guinea C strain, *AF038403.1*) was purchased from American Type Culture Collection (ATCC) (VR-1584, ATCC). DENV-2 (43 strain, *AF204178*), JEV (P3 strain, *U47032.1*), SINV (*U90536.1*), SFV (*NC_003215.1*), BATV (YN92-4 strain)[65], TAHV (XJ0625 strain, *EU622820.1*), and ZIKV (GZ01 strain,

*KU820898.1*) were provided by National Institute for Viral Disease Control and Prevention, Chinese Center for Viral Disease Control and Prevention. The viruses were passaged in C6/36 cells for mosquito microinjection. The viruses were grown in Vero cells in VP-SFM serum-free medium (11681-020, Gibco) for the blood meal. The Vero (CCL-81, ATCC) and C6/36 (CRL-1660, ATCC) cells were purchased from ATCC. The C6/36 and Vero cells were maintained in 10% heat-inactivated fetal bovine serum (16000-044, Gibco). The virus titers were determined by a plaque formation assay and the 50% mosquito infectious dose (M.I. D.$_{50}$), as described previously[49, 50, 63].

**Gene silencing in mosquitoes**. Detailed procedures for gene silencing in mosquitoes have been described elsewhere[49]. Briefly, female mosquitoes were anaesthetized on a cold tray and 1 μg/300 nl of dsRNA was microinjected into their thoraxes. The injected mosquitoes were allowed to recover for 3 days under standard rearing conditions. They were subsequently used for infection studies. For infection by thoracic microinjection, 10 M.I.D$_{50}$/300 nl virus was inoculated into per mosquito. The gene silencing efficiency was assessed by qPCR. The primers used for gene detection are shown in Supplementary Data 2.

**Mosquito in vitro membrane feeding**. Fresh human blood was collected in heparin-coated tubes (367884, BD Vacutainer) and centrifuged at 1000×*g* and 4 °C for 10 mins to separate plasma from blood cells. The plasma was collected and heat-inactivated at 55 °C for 60 mins. The separated blood cells were washed three times with PBS to remove the anticoagulant. The cells were then re-suspended in the heat-inactivated plasma. Next, 1% sucrose was filtered to eliminate bacteria. The materials were mixed with 1% sucrose or human blood for mosquito oral feeding via a Hemotek system (6W1, Hemotek). $6 \times 10^5$ p.f.u./ml of DENV-2, $5 \times 10^5$ p.f.u./ml of JEV, $2 \times 10^6$ p.f.u./ml of SINV, $5 \times 10^6$ p.f.u./ml of TAHV and $5 \times 10^5$ p.f.u./ml of ZIKV were used for mosquito oral infection. Fully engorged female mosquitoes were transferred into new containers and maintained under standard conditions for additional days. The mosquitoes were subsequently sacrificed for further investigation.

**Quantitation analysis of viral genome by qPCR**. Total RNA was isolated from homogenized mosquitoes using an RNeasy Mini Kit (74106, Qiagen) and reverse transcribed into complementary DNA (cDNA) using an iScript cDNA synthesis kit (170-8890, Bio-Rad). Viral genomes were quantified by SYBR green qPCR or TaqMan qPCR amplifications. The primers and probes used for these analyses are shown in Supplementary Data 2. Gene quantities were normalized against *A. aegypti* and *C. pipiens pallens* actins.

**RNA-seq of mosquito genes**. Total RNA was extracted with TRIzol (Ambion Cat. No# 15596018) from the mosquitoes that were microinjected with 100 M.I.D.$_{50}$ viruses, respectively. The samples were delivered to the Beijing Genomics Institute (Shenzhen, China) for commercial RNA-Seq services and data analysis. Clean reads were mapped to the *A. aegypti* transcript database using SOAPaligner/SOAP2 mismatches. The number of clean reads for each gene was calculated and then normalized to Reads per Kb per Million reads, which associates read numbers with gene expression levels. The log$_2$ ratio (read number in virus-induced whole mosquito/read number in control whole mosquito) was exploited to evaluate gene regulation. Genes with a log$_2$ ratio $\leq -1.5$ or $\geq 1.5$ were selected for further analysis. For analysis of the *AaGABA$_A$-R1* silencing in mosquitoes, total RNA was extracted with TRIzol (Ambion Cat. No# 15596018) from the *GFP* dsRNA and *AaGABA$_A$-R1* dsRNAs inoculated mosquitoes at 3 days post dsRNA treatment. Immune genes with a log$_2$ ratio $\leq -0.5$ or $\geq 0.5$ were selected for further analysis. A rigorous algorithm was used to screen for differentially expressed genes in each group. The sequencing data were deposited in the Short Read Archive (NCBI) under accession number GSE91027 and GSE91032.

**Detection of GABA and glutamic acid in mosquitoes**. Protein lysates in PBS were collected from whole mosquitoes over time after the mosquito meal. The concentration of GABA was quantified using a GABA ELISA kit (BA E-2500, LDN). The amount of glutamate was measured with a glutamate assay kit (MAK004-1KT, SIGMA). The experiment was performed according to the manufacturer's manual. The optical density was measured at 450 nm with a multimode reader (Varioskan Flash Multimode Reader, Thermo Scientific).

**Insecticide assays**. The detailed experimental procedures are described in the WHO document, 'Supplies for monitoring insecticide resistance in diseases vectors' (WHO). Briefly, a clean white paper ($6 \times 6$ cm$^2$) rolled into a cylinder shaper and impregnated either with Fipronil (50 μg/bottle), Bilobalide (100 μg/bottle) or Deltamethrin (50 ng/bottle) was inserted into a 100 ml bottle (14395-100, Kimble). Thirty to thirty-five female mosquitoes, infected by either DENV-2 or ZIKV by oral feeding, were transferred are exposed in these bottle for 2 h. Surviving mosquitoes were transferred to new culture containers for further investigation.

**Vector competence assay**. A mixture, which contained 1% sucrose (50% v/v), supernatant from infected Vero cells (cultured in serum-free medium) (50% v/v),

and 100 μg/ml glutamic acid, was used to feed mosquitoes via an in vitro blood feeding system. Mosquitoes feeding on the mixture without glutamic acid served as negative controls. The midguts, heads, and salivary glands were dissected on 7, 14, and 14 days after oral feeding, respectively. The viral loads were measured by qPCR. The ratio of midgut infection, dissemination and transmission were calculated by Positive Number/Total Number of midguts, heads and salivary glands, respectively.

**Statistics**. Mosquitoes that died before measurement were excluded from the analysis. The investigators were not blinded to the allocation during the experiments or to the outcome assessment. Descriptive statistics have been provided in the figure legends. Given the nature of the experiments and the type of samples, differences in continuous variables were assessed with the non-parametric Mann–Whitney test. Differences in mosquito infective rates were analyzed by Fisher's exact test. All analyses were performed using GraphPad Prism statistical software.

**Data availability**. The sequencing data were deposited in the Short Read Archive (NCBI) under accession number GSE91027 and GSE91032. All other data that support the study are available from the corresponding author upon reasonable request.

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

## Acknowledgements

This work was funded by the grants from the National Key Research and Development Plan of China (2016YFC1201000, 2017ZX09101-005, 2016YFD0500400, 2016ZX10004001-008), the grants from the National Natural Science Foundation of China (81730063, 81422028, 81571975, and 81290342), and the grant from National Institute of Health of the United States (AI103807), and Shenzhen San-Ming Project for prevention and research on vector-borne diseases. G.C. is a Newton Advanced Fellow awarded by the Academy of Medical Sciences and the Newton Fund. We thank the technical supports from the Core Facility of Center for Life Sciences and Center of Biomedical Analysis (Tsinghua University).

## Author contributions

G.C. designed the experiments and wrote the manuscript; Y.Z. performed the majority of the experiments and analyzed data; R.Z. and B.Z. helped with the RNA isolation and qPCR detection; T.Z. and G.L. provided experimental materials for the investigation. T.Z., G.L. and P.W. contributed experimental suggestions and strengthened the writing of manuscript. All authors reviewed, critiqued, and provided comments to the text.

## Additional information

**Competing interests:** The authors declare no competing financial interests.

