## [Peer Review File · Nature Communications]

Reviewers' comments:

Reviewer #1 (Remarks to the Author):

Zhu et al describe the effects of bloodmeal and GABA signalling on arbovirus infection of mosquitoes. Overall these are novel and interesting insights, though I think the title of the manuscript could be more precise.

The key conclusion is that infection is facilitated by the GABAergic system as this inhibits antiviral immunity thus supporting infection and replication; this is done in *Ae. aegypti* with a range of arboviruses representing all major families and is at the outset a RNA seq analysis at several days post infection. Intriguingly 19 upregulated genes were shared between all families and gene by gene silencing approach with dengue virus which identified the GABA receptor subunit R1 as having proviral activity that extended to togaviruses and bunyaviruses. I liked the experiments with insecticide drugs (survival rates were assessed) that inhibit these receptors and AaGAD1 silencing as they nicely back up the earlier silencing experiments and consolidate the broader importance of the findings. It is interesting that this activity appears to target the Imd pathway specifically, and this novel to the field. It would be very useful to provide some data on how GABA may affect the IMD pathway. Is glutamic acid directly regulating its activity, for example did feeding this amino acid affect its activity and proliferation of gut bacteria? This is key point of the paper and should be explored further.

Overall I found the manuscript clear and well explained, but it should be read by a native English to work on some minor inconsistencies.

Comments:

- 1) Line 55: "Intriguingly, mosquitoes are very permissive to these viruses through a blood meal, regardless of the dramatic difference in their infection properties." I don't think that statement is correct or clear; moreover there are differences between *Ae. aegypti* strains with regards to competence. I would be careful with such broad statements.
- 2) Virus names are incorrectly spelled at times. Geographical locations are in capital letters, for example Semliki Forest virus not Semliki forest virus, Japanese encephalitis virus etc.
- 3) Are the infection rates with all these viruses similar? The original analysis was carried out by intrathoracic injection so I would assume good infection rates but I wonder to what extent the authors checked.
- 4) Could insecticides have additional effects as inhibitors of arbovirus infection?
- 5) The inclusion of JEV and *Culex pipiens* is interesting but it would have been more consistent to do the bloodmeal experiments also with bunyaviruses.
- 6) I like that the authors prepared viruses for bloodmeal in vertebrate cells and those for microinjection in C6/36 cells. Give the different properties of viruses produced in either cell,

and looking at two very different biological situations, makes this a strong point which in my opinion has been neglected in many studies in this field.

Reviewer #2 (Remarks to the Author):

General comments:

This manuscript presents a data-rich study based on a comprehensive approach of functional genomics to identify molecular mechanisms underlying mosquito susceptibility to arbovirus infection. The study addresses an important question and may have discovered a key mechanism regulating arbovirus infection of mosquitoes – namely the GABAergic system. However, the study in its present form seems premature for publication. This is because, while several interesting correlations were observed, the causal relationships were not fully established. Overall the manuscript (and its title) is misleading because it claims to have established a functional link between blood feeding and arbovirus infection that has not been conclusively demonstrated.

Major comments:

Put in a conceptual framework, an effect of A on B, an effect of B on C, and an effect of A on C, together do not necessarily imply that the effect of A on C is mediated by B. This shortcoming applies to two of the main conclusions put forward in the manuscript: (1) the proviral effect of the GABAergic system on arbovirus infection is mediated by suppression of the Imd pathway (lines 219-220, lines 288-290); (2) the proviral effect of blood feeding (or protein ingestion) on arbovirus infection is mediated by activation of the GABAergic system (lines 222-223). Using the theoretical notations introduced above, in both cases the authors showed that B was necessary but they did not demonstrate that it was sufficient to explain the effect of A on C.

An initial transcriptomic analysis was performed 6 days after intrathoracic virus inoculations (and therefore had nothing to do with blood feeding). This initial analysis identified a potential role for the GABAergic system in arbovirus infection, which was confirmed by functional assays. Second, a subsequent transcriptomic analysis showed that impairment of the GABAergic system leads to the upregulation of the Imd pathway. Functional assays then confirmed the previously known antiviral role of the Imd pathway during arbovirus infection, however there was no evidence for a causal relationship between the proviral effect of the GABAergic system and suppression of the Imd pathway. Third, the study examined the effect of blood feeding (or protein ingestion) on glutamate and GABA concentration, and on arbovirus infection. This part of the study showed, in separate experiments, that (i) blood feeding activates the GABAergic system, (ii) ingestion of glutamic acid facilitates arbovirus infection, and (iii) silencing the GABAergic system inhibits arbovirus infection.

Additional experiments are required to complete the demonstration. For example, the claim that the GABAergic systems facilitates arbovirus infection via suppression of the Imd

pathway would be significantly strengthened by an experiment showing that the effect of AaGABAA-R1 knockdown is rescued by Imd overactivation. Likewise, combining the blood feeding experiment shown in Fig. 5AB with AaGAD1 knockdown like in Fig. 6DE would support the claim that blood feeding promotes arbovirus infection via activation of the GABAergic system.

Another main concern is about the insecticide experiments (Fig. 2C-F). In these experiments the mosquitoes were exposed to two different insecticides that are inhibitors of GABA receptors. However, using unexposed mosquitoes as the control treatment means that the experiment is essentially testing the effect of insecticide exposure, not the effect of GABA signalling blockade (as claimed on line 158), on arbovirus infection. The proper control should have consisted of mosquitoes exposed to an insecticide that does not target GABA signalling.

On several occasions the results of the study were extrapolated and put in the context of arbovirus transmission by mosquitoes and human vector-borne diseases. The authors should use more caution given the very artificial nature of their experimental setup. First, the *Aedes aegypti* Rockefeller strain is a very old, lab-adapted strain, and so are some of the virus strains used in the study. Second, a large fraction of mosquito experimental infections are performed by intrathoracic inoculation, which bypasses the midgut barrier and is an unnatural mode of infection. Third, the main phenotypic readout (DENV E / Actin mRNA ratio) measures the concentration of viral RNA in the whole mosquito body but does not directly relate with conventional vector competence phenotypes (midgut infection, viral dissemination, and transmission). Moreover, this readout is not equivalent to the amount of infectious viral particles and could be misleading. It would be useful to validate some of the key results with an infectious assay such as plaque assay.

The abstract, introduction and discussion sections include adaptive evolutionary interpretations that do not make sense. For example, "mechanism employed by mosquitoes to efficiently acquire viruses" (lines 38-39), "mosquitoes have developed some mechanisms that are common to all viruses to facilitate viral infections" (lines 55-56), "a co-evolutionary relationship between mosquito-borne viruses and their vectors, allowing mosquitoes to robustly carry and transmit the viruses" (lines 302-303), give the impression that mosquitoes adapted to become vectors of viruses during evolution. It is unclear (and even counter-intuitive) why mosquitoes would have been selected to become susceptible to viruses, or even why they would have co-evolved with arboviruses.

Minor comments:

Several figure panels are largely redundant: Fig. 5A and Fig. 5B, Fig. 5C and Fig. 5D, Fig. 6A and Fig. 6B, Fig. 6D and Fig. 6E, Fig. 6F and Fig. 6G, Fig. 6H and Fig. 6I.

The information presented in Fig. 1A is unclear. What does this panel show?

Survival data should be provided for the GABA injection experiments.

Line 45: "a habit of mosquitoes" should be rephrased as "a behavioral trait of mosquitoes".

Line 56: "common to all viruses" should be rephrased as "common to all mosquito-borne human viruses".

Line 92, line 291: According to the American CDC, there are only about a hundred of known arboviruses that are pathogenic to humans.

Line 95: "dengue" and "encephalitis" should not be capitalized.

Line 102: The number of genes, not the genes.

Line 151: "We then infected those surviving mosquitoes with DENV-2". This statement is contradictory to what is specified in the legend of Fig. 2CD, which indicates that virus and insecticide were simultaneously inoculated. Please clarify.

Line 162: What was the mode of ZIKV infection? Please specify.

Line 196: Why were two different dsRNAs used? Please justify.

Lines 225-226: "the blood meal is an approach by which mosquitoes acquire arboviruses from infected hosts". Actually, this is the primary mode of mosquito infection by arboviruses.

Line 251: Please specify the infectious dose of virus used to orally infect mosquitoes in Fig. 6, to improve accuracy and allow reproducibility.

Line 374-377: It is unclear whether or not 300 nL of liquid was inoculated into the mosquitoes, or if "1 ug per 300 nL" is a concentration. Please correct "injection into virus" on line 377.

Line 391: "killed" should be replaced by "sacrificed".

Line 440: "Flies" must be a typo.

Reviewer #3 (Remarks to the Author):

The manuscript, 'The effects of blood meal acquisition on infection of mosquitoes with arboviruses', by Zhu et al., describes experiments which identified activation of the GABAergic system as one factor facilitating competence of mosquitoes for arboviruses. The mechanism of this facilitation was found to be suppression of the Imd immune pathway. GABA activation was also found to be associated with blood meal acquisition/digestion, as a result of protein acquisition/digestion, specifically glutamic acid activation of GABA. Evidence of the effect of GABA on arbovirus competence was found across viral and mosquito species, thereby permitting the authors to conclude that this was a conserved

phenomenon. A particularly interesting supplemental finding was that treatment with some common insecticides inhibit GABA and subsequently virus load. This is novel and interesting work which is well-designed and generally well presented. Many of the conclusions are well supported, yet there are some significant concerns with some experimental details and interpretation, particularly those utilizing sugar meals. In addition, the magnitude of the findings and the implications for vector/virus evolution are overstated. Specific critiques follow:

General concerns:

1. Title and related claims- 'the effects of blood meal acquisition on infection of mosquitoes' are not actually comprehensively studied. To do this, the investigators would need to assure that viral doses to gut epithelial cells acquired through sugar feeding and blood meal feeding were similar, determine the full extent of genes up or downregulated by blood meal digestion and more comprehensively establish the role of each potentially altered gene/pathway on vector competence. Instead, a target was identified through well-supported hypotheses and inoculation studies and the effect of altering that pathway exclusively was studied with blood meal feeding experiments. The title and related language should be changed. Perhaps, 'Blood meal acquisition enhances arbovirus replication in mosquitoes through activation of the GABAergic system'. ?
2. The findings are presented as a discovery of the primary mechanism resulting in permissiveness of mosquitoes to arboviruses. There is no experimental result demonstrating that the absence of GABA completely ablates midgut infection, viral replication or dissemination/transmission (not studied). This result makes sense since Imd, although it clearly has a secondary role, is not the primary immune pathway for defense against RNA viruses. Since GABA has no effect on expression of any molecule which has been implicated in the RNAi pathway, it is not independently capable of facilitating permissiveness to RNA virus infection. Published evidence suggests that this is accomplished by specific viral sequences/proteins evading, suppressing or sequestering the RNAi response, in addition to other mechanisms. There is no mention of the fact that the Imd pathway is primarily used as a defense against Gram negative bacteria. There are multiple statements that seem to suggest that the mosquitoes evolved this system to allow the pathogen to efficiently infect (see line 278). This does not make evolutionary sense. First, even during the most severe outbreaks arboviruses are not prevalent enough to influence mosquito evolution, and second, there is no documented fitness advantage to arbovirus infection in mosquitoes that would drive this. It makes more sense that the pathogen is simply exploiting a system that has evolved for other purposes (perhaps to protect against costly immune activation in the presence of bacteria laden blood). The discussion should be reworked so that the findings are not overstated and the implications fully consider what is known about these systems.
3. Doses- Unless I am missing it, there do not seem to be any reports of the doses used to infected mosquitoes in feeding experiments. This is critical information for interpretation.
4. No assessment of competence- Only fold changes in viral load and proportion infected are compared so the impact on transmission is not clear.

Specific critiques (in order of appearance):

1. line 44- In the intro and discussion blood feeding is referred to as a 'common property' of hematophagous insects. That seems to imply that it just happens frequently rather than it being the defining characteristic of these insects.

2. Line 52- Add the word 'Most'. There are pathogenic viruses from other families transmitted by mosquitoes.
3. Line 54- 'Mosquitoes are very permissive' seems to ignore the fact that many are highly refractory and that specific species are responsible for transmission of specific viruses.
4. Line 84- 'facilitating effective systemic viral infection' is an overstatement. Infection is still established in many mosquitoes and, although it would significantly strengthen the significance, dissemination and transmission were not assessed.
5. Line 65-6- Dengue and Encephalitis should not be capitalized.
6. Line 95-100- It is worth noting that only two of the viruses used are transmitted by *Ae. aegypti* in nature, while 3 are vectored by *Culex* mosquitoes and one (Batai) by *Anopheles*. These are really important differences in the interpretation of results. For instance, in figure 1, gene regulation in *Ae. aegypti* is being compared among viruses which have completely different competence levels in this species. The highest number of genes for which varied expression was measured are with the combination that is most frequent in nature (DENV-*aegypti*). Somewhat related to this, were virus levels quantified on days 1 and 6? It would be useful to know how viral titer/replication relates to level/spectrum of expression.
7. Lines 119-21/suppl fig. 1- There is no further discussion/study of the 2 genes identified as potentially important in suppressing arbovirus infection (i.e. when silenced DENV replication increases). Would seem to be warranted.
8. Figure 2 A/B- It is convincing that the titer drop equates to a drop in GABA and this is a sound result. Again, though, authors need to be careful with the biological significance when interpreting. While it is not totally clear what the level of detection is here and how fluorescence equates to viral titer, a 2-4 fold drop is quite small in systems in which titer are normally evaluated on a log₁₀ scale. Similar differences were measured when silencing other genes.
9. Lines 230-1- Viruses do not simultaneously infect the gut and spread into the hemocoel as stated. Dissemination is delayed until significant replication in the gut is achieved and often not attained at all due to the midgut escape barrier. Please restate.
10. Figure 5- Mosquitoes are very unlikely to become fully engorged with a sugar meal as they do with a blood meal. For this reason, it is almost certain that in the sugar meal experiments the viral dose was lower. This could wholly explain the differences that were measured. This needs to be addressed, possibly by quantifying virus at time 0 (following feeding) and possibly adjusting concentrations so that mosquitoes receive similar doses.
11. Lines 250-2- Sentences beginning on line 250 and 251 seem to contradict. All suppl. Fig 8 shows is that glutamic acid doesn't affect the ability of DENV to form plaques on Vero cell culture. This may or may not tell you anything about infectiousness in vivo in mosquito gut epithelial cells.
12. Line 284-Remove the word 'dissemination'. This is not evaluated in the study.
13. Line 294-Again, mosquitoes don't 'exploit' these receptors, pathogens exploit them. This seems to be consistent with many statements suggesting there is an evolutionary advantage for the mosquito to be permissive to arboviruses.

Responses to Referee #1:

#1A. *It would be very useful to provide some data on how GABA may affect the IMD pathway. Is glutamic acid directly regulating its activity, for example did feeding this amino acid affect its activity and proliferation of gut bacteria? This is key point of the paper and should be explored further.*

#1A Answer: We agree with the reviewer's suggestion. In this study, we found that the activation of the GABAergic system significantly reduces the expression of many key components of the Imd signaling pathway in the *Aedes aegypti* midgut. Glutamic acid-derived GABA, generated by blood digestion, suppresses the Imd pathway-mediated immune responses by enhancing the GABA signaling. To address the reviewer's concern, we first assessed whether feeding GABA may affect the Imd pathway. Since proliferation of the microbiota after a meal can skew the Imd signaling in the midgut, we removed the gut commensal microbiome by feeding antibiotics to the mosquitoes (Pang et al., 2016, Nature Microbiology, 1, 16023). The oral introduction of 120 ng of GABA with sucrose reduced the expression of the *AaImd*, *AaRel2*, *AaDef-C* and *AaCec-N* genes in the midgut of antibiotic-treated mosquitoes (Supplementary Figure 10A-D). The abundance of the gut microbial flora is an indicator of local immune activity (Pang et al., 2016, Nature Microbiology, 1, 16023; Xiao et al., 2017, Nature Microbiology, 1, 17020). Sucrose feeding with 120 ng of

GABA enhanced the burden of commensal bacteria in the mosquito guts (Supplementary Figure 10E) (Page 9, Line 227-232).

In the original experimental conditions, we could not rule out the possibility that the glutamic acid generated by blood meals may directly regulate the activity of Imd signaling. To address the reviewer's concern, we impaired the GABAergic system by the dsRNA-mediated knockdown of *Glutamic Acid Decarboxylase* gene (*AaGAD*) in *A. aegypti*. Mosquitoes treated with *Green Fluorescent Protein (GFP)* dsRNA served as negative controls. We removed the gut microbiome by feeding antibiotics. Six days post dsRNA inoculation, the antibiotic-treated mosquitoes were fed by 1% sucrose with or without 100 µg/ml glutamic acid. The expression of Imd component genes were determined at 18 hours post sucrose feeding. The oral introduction of glutamine impaired the expression of the *AaImd*, *AaRel2*, *AaDef-C* and *AaCec-N* genes in the midgut of antibiotic-treated mosquitoes. However, silencing the *AaGAD* gene offset these expression impairments (Supplementary Figure 12A-D). These result indicate that glutamic acid does not directly regulate the Imd signaling activity, but rather impairs the pathway through GABAergic-mediated GABA generation. Subsequently, we assessed whether feeding glutamic acid may affect the microbial proliferation in the mosquito guts. After a sucrose meal with glutamic acid, the burden of commensal bacteria was enhanced in the mosquito gut. However, knockdown of the *AaGAD* gene rescued the microbial proliferation caused by glutamic acid introduction (Supplementary Figure 12E), further indicating glutamic acid modulates the gut immune activity through the GABAergic system. We have added these results into the revised manuscript (Page 11, Line 283-285).

#1B. *Line 55: "Intriguingly, mosquitoes are very permissive to these viruses through a blood meal, regardless of the dramatic difference in their infection properties." I don't think that statement is correct or clear; moreover there are differences between Ae. aegypti strains with regards to competence. I would be careful with such broad statements.*

#1B Answer: We recognize the reviewer's concern. We have deleted this sentence and revised the statement to *"Since mosquitoes are primary vectors for the transmission of these viruses, we speculate that arboviruses may exploit some common mechanisms to facilitate their infections in mosquitoes"* (Page 3, Line 56-58).

#1C. *Virus names are incorrectly spelled at times. Geographical locations are in capital letters, for example Semliki Forest virus not Semliki forest virus, Japanese encephalitis virus etc.*

#1C Answer: Thank you for the corrections. We have revised the virus names in the manuscript (Page 5, Line 101; Page 17, Line 465).

#1D. *Are the infection rates with all these viruses similar? The original analysis was carried out by intrathoracic injection so I would assume good infection rates but I wonder to what extent the authors checked.*

#1D Answer: Based on the reviewer's suggestion, we determined the replication rates of Dengue (DENV), Japanese Encephalitis (JEV), Semliki Forest (SFV), Sindbis (SINV), Tahyna (TAHV) and Batai (BATV) viruses in *A. aegypti* mosquitoes infected via intrathoracic microinjection. We used 100 M.I.D.₅₀ (50% Mosquito Infective Dose) each virus to infect the female mosquitoes. The virus burdens in mosquitoes were quantified by qPCR over a time course post-infection. For the DENV, JEV, TAHV and BATV infections, the viral loads were low on 1 days post-infection (dpi) and subsequently increased from 3 to 9 dpi (Supplementary Figure 1). However, both of the alphaviruses, SINV and SFV, replicated robustly from 1 to 3 dpi, and their viral burdens were then saturated in viral burden after 3 dpi (Supplementary Figure 1), suggesting that alphaviruses may more aggressively infect and disseminate in the *A. aegypti* mosquitoes (Richard et al., 2016, PLoS Negl Trop Dis, 10: e0004694). Based on the infection rates of the different viruses in the mosquitoes, we analyzed gene regulation on days 1 and 6 of the infection, which represent early and late infection in the mosquitoes, respectively. (Page 5, Line 104-107)

#1E. Could insecticides have additional effects as inhibitors of arbovirus infection?

#1E Answer: We recognize the reviewer's concern. The original results have showed that replication of DENV in mosquitoes was impaired by either thoracal inoculation or container spraying of Fipronil and Bilobalide. To test whether both of the insecticides have additional effects on arbovirus infection, we first premixed DENV-2 (the supernatant from infected Vero cells) with either 50 µg of Fipronil or 100 µg of Bilobalide. Virus incubated with PBS served as negative controls. Thirty

Figure R1. Incubation of Fipronil and Bilobalide did not affect the DENV-2 infectivity. 6×10^5 p.f.u. ml⁻¹ DENV-2 was premixed with either 50 µg Fipronil or 100 µg Bilobalide for 30 mins, respectively. The virus incubated with PBS served as negative control. The viral titer was determined by a plaque assay. The experiment was reproduced by two times.

minutes post-incubation, the viral titers were determined by a plaque assays. Incubation with these insecticides did not affect the DENV-2 infectivity (Figure R1).

Both of the insecticides targets the AaGABA_A receptor. We proposed that the insecticide-mediated impairment of DENV replication might be the result of the inhibition of mosquito GABA signaling. We therefore assessed whether the insecticides have additional effects on arbovirus infection in GABA signaling-deficient mosquitoes. Knockdown of AaGABA_A-R1 reduced the DENV load in the mosquitoes. However, inoculation with either 15 ng of Fipronil or 30 ng of Bilobalide per mosquito did not show any additional effect on the DENV-2 replication in the AaGABA_A-R1-silenced mosquitoes (Supplementary Figure 6), suggesting that neither of these insecticides have additional effects on arbovirus infection in mosquitoes (Page 7, Line 166-169).

#1F. *The inclusion of JEV and Culex pipiens is interesting but it would have been more consistent to do the bloodmeal experiments also with bunyaviruses.*

#1F Answer: Thank you for the suggestion. Tahyna virus (TAHV), transmitted by *Culex* mosquitoes in nature (Lu et al., 2009, Emerg Infect Dis, 10: 130), is a typical bunyavirus belonging to the *Orthobunyavirus* genus. We therefore exploited TAHV and *Culex pipiens pallens* to assess the effect of blood meals on arbovirus infection. Consistently, feeding blood largely enhanced the TAHV infection in *C. pipiens pallens*, compared to that of sucrose ingestion (Figure 5C).

Blood meals may result in the enhancement of GABA signaling because a large amount of glutamic acid is produced from blood digestion. Therefore, we assessed whether the oral introduction of glutamic acid could enhance TAHV infection of *C. pipiens pallens*. The oral introduction of glutamic acid by a sucrose meal enhanced TAHV infection of the *Culex* mosquitoes (Figure 6F and Supplementary Figure 14D), further indicating that mosquito hematophagy can enhance GABAergic signaling to commonly facilitate arbovirus infection (Page 10, Line 259-262; Page12, Line 320-323).

#1G. *I like that the authors prepared viruses for bloodmeal in vertebrate cells and those for microinjection in C6/36 cells. Give the different properties of viruses produced in either cell, and looking at two very different biological situations, makes this a strong point which in my opinion has been neglected in many studies in this field.*

#1G Answer: Thank you for the comments.

Responses to Referee #2:

#2A. *Put in a conceptual framework, an effect of A on B, an effect of B on C, and an effect of A on C, together do not necessarily imply that the effect of A on C is mediated by B. This shortcoming applies to two of the main conclusions put forward in the manuscript: (1) the proviral effect of the GABAergic system on arbovirus infection is mediated by suppression of the Imd pathway (lines 219-220, lines 288-290); (2) the proviral effect of blood feeding (or protein ingestion) on arbovirus infection is mediated by activation of the GABAergic system (lines 222-223). Using the theoretical notations introduced above, in both cases the authors showed that B was necessary but they did not demonstrate that it was sufficient to explain the effect of A on C.*

#2A Answer: We agree with the reviewer's concern regarding the conceptual framework. Based on our understanding, *A* represents the effect of blood feeding; *B* is the activation of the GABAergic system; and *C* is the regulation of arboviral replication in mosquitoes. The original data has showed that the GABAergic signaling was stimulated by mosquito blood meals (the effect of *A* on *B*) (Figure 5E) and that the activation of the GABAergic system facilitated arbovirus infection in mosquitoes (the effect of *B* on *C*) (Figure 2G and Supplementary Figure 8). Furthermore, the arbovirus infective ratios of the mosquitoes were significantly enhanced by feeding blood (the effect of *A* on *C*) (Figure 5A-C). To assess whether GABAergic signaling (*B*) might

play a role in the effect of blood feeding (A) on arbovirus infection (C), we interrupted the GABAergic system by *AaGAD1* knockdown. Silencing *AaGAD1* partially offset the enhancing effect of blood feeding on DENV infection (Supplementary Figure 11), indicating that the GABAergic system (B) contributes to the enhancement of arbovirus infection (C) by blood meals (A). We have added this result in the revised manuscript (Page 10, Line 262-264).

#2B. *An initial transcriptomic analysis was performed 6 days after intrathoracic virus inoculations (and therefore had nothing to do with blood feeding). This initial analysis identified a potential role for the GABAergic system in arbovirus infection, which was confirmed by functional assays. Second, a subsequent transcriptomic analysis showed that impairment of the GABAergic system leads to the upregulation of the Imd pathway. Functional assays then confirmed the previously known antiviral role of the Imd pathway during arbovirus infection, however there was no evidence for a causal relationship between the proviral effect of the GABAergic system and suppression of the Imd pathway. Third, the study examined the effect of blood feeding (or protein ingestion) on glutamate and GABA concentration, and on arbovirus infection. This part of the study showed, in separate experiments, that (i) blood feeding activates the GABAergic system, (ii) ingestion of glutamic acid facilitates arbovirus infection, and (iii) silencing the GABAergic system inhibits arbovirus infection. Additional experiments are required to complete the demonstration. For example, the claim that the GABAergic systems facilitates arbovirus infection via suppression of the Imd pathway would be significantly strengthened by an experiment showing that the effect of *AaGABA_A-R1* knockdown is rescued by Imd overactivation.*

#2B Answer: We agree with the reviewer's concern that more evidence is needed to claim a relationship between the GABAergic system, the Imd pathway and arbovirus infection. The original data have indicated that (i) inoculation of GABA enhances DENV replication in mosquitoes, (ii) the GABAergic system negatively regulates the Imd signaling response and (iii) the Imd pathway plays a vital role in the immune response against DENV infection in mosquitoes. We first assessed whether activation of the Imd response could offset the GABA-mediated enhancement of DENV infection. Inoculation of GABA significantly enhanced the DENV-2 infection in mosquitoes (Figure 4L), which is consistent with the previous observation (Figure 2G). However, the activation of Imd signaling by peptidoglycan (PGN) inoculation abolished the enhancing effect of GABA on DENV-2 infection (Figure 4L). Furthermore, interruption of the Imd pathway increased DENV-2 infection in the *AaGABA_A-R1*-silenced mosquitoes to that of *GFP* RNAi-treated mosquitoes (Figure 4M). These data clearly suggest a linear GABA-*AaGABA_A-R1*-Imd axis (Page 10, Line 241-245).

We next assessed whether feeding GABA can affect the Imd pathway. Since the proliferation of microbiota after a meal can skew the Imd signaling in the midgut, we removed the gut commensal microbiome by feeding antibiotics in mosquitoes (Pang et al., 2016, *Nature Microbiology*, 1, 16023). The oral introduction of 120 ng of GABA with sucrose reduced the expression of the *Aalmd*, *AaRel2*, *AaDef-C* and *AaCec-N* genes in the midgut of antibiotic-treated mosquitoes (Supplementary Figure 10A-D).

The abundance of the gut microbial flora is an indicator of local immune activity (Pang et al., 2016, *Nature Microbiology*, 1, 16023; Xiao et al., 2017, *Nature Microbiology*, 1, 17020). Sucrose feeding with 120 ng of GABA enhanced the burden of commensal bacteria in the mosquito guts (Supplementary Figure 10E). Taken together, both of these functional experiments validated that the GABAergic systems may facilitate arbovirus infection via suppression of the Imd pathway (Page 9, Line 227-232).

#2C. *Likewise, combining the blood feeding experiment shown in Fig. 5AB with AaGAD1 knockdown like in Fig. 6DE would support the claim that blood feeding promotes arbovirus infection via activation of the GABAergic system.*

#2C Answer: The original results indicated that feeding blood largely enhanced the infection of DENV-2 in *A. aegypti* compared to that of sucrose meals. To further investigate whether the GABAergic system plays a role in the promotion of arbovirus infection through blood meals, we interrupted the GABAergic system by knockdown of the *AaGAD1* gene. *GFP* dsRNA-inoculated mosquitoes served as negative controls. The mosquitoes from both of groups were orally infected with DENV-2 via blood meal. Silencing *AaGAD1* partially offset the enhancement of DENV infectivity caused by blood feeding (Supplementary Figure 11), indicating that the GABAergic system partially contributes to the promotion of arbovirus infection through blood meals (Page 10, Line 262-264).

#2D. *Another main concern is about the insecticide experiments (Fig. 2C-F). In these experiments the mosquitoes were exposed to two different insecticides that are inhibitors of GABA receptors. However, using unexposed mosquitoes as the control treatment means that the experiment is essentially testing the effect of insecticide exposure, not the effect of GABA signalling blockade (as claimed on line 158), on arbovirus infection. The proper control should have consisted of mosquitoes exposed to an insecticide that does not target GABA signalling.*

#2D Answer: We recognize the reviewer's concern. To improve the experiment, we selected Deltamethrin as a control insecticide that does not target the GABAergic system (Zeng et al., 2017, *Pestic Biochem Physiol*, 138: 84-90). Thoracic inoculation of Deltamethrin, Fipronil and Bilobalide all caused mosquito death in an insecticide-dose dependent manner (Supplementary Figure 5). Compared to the significant reduction in viral replication with both Fipronil and Bilobalide-inoculated mosquitoes, treatment with Deltamethrin did not influence the DENV-2 burden in *A. aegypti* (Figure 2D-E). To validate the phenotypes in relatively natural settings, *A. aegypti* mosquitoes that were infected with DENV-2 orally via a blood meal were maintained in the containers sprayed with either Fipronil (50 µg/bottle), Bilobalide (100 µg/bottle) or Deltamethrin (50 ng/bottle), respectively. Exposure to both Fipronil and Bilobalide (Figure 2F), and not Deltamethrin, reduced the DENV burden in the mosquitoes that survived over the time course of insecticide treatment, further validating that the blockage of GABA signaling by these insecticides reduced arbovirus replication in mosquitoes (Page 7, Line 169-175).

#2E. *On several occasions the results of the study were extrapolated and put in the context of arbovirus transmission by mosquitoes and human vector-borne diseases. The authors should use more caution given the very artificial nature of their experimental setup. First, the Aedes aegypti Rockefeller strain is a very old, lab-adapted strain, and so are some of the virus strains used in the study. Second, a large fraction of mosquito experimental infections is performed by intrathoracic inoculation, which bypasses the midgut barrier and is an unnatural mode of infection. Third, the main phenotypic readout (DENV E / Actin mRNA ratio) measures the concentration of viral RNA in the whole mosquito body but does not directly relate with conventional vector competence phenotypes (midgut infection, viral dissemination, and transmission). Moreover, this readout is not equivalent to the amount of infectious viral particles and could be misleading. It would be useful to validate some of the key results with an infectious assay such as plaque assay.*

#2E Answer: The *Aedes aegypti* Rockefeller strain was established as a lab-adapted strain in 1930 (Kuno et al., 2010, J Med Entomol, 47: 957-971) and is widely used as one of the standard *A. aegypti* strains for the investigation of mosquito-borne viral infections (Araújo et al., 2013, Parasite Vectors, 6: 297; Londono-Renteria et al, 2015, PloS Pathog, 11: e1005202). The DENV-2 strain used in this study was the New Guinea C strain (AF038403.1), which was isolated in 1944 (Irie et al., 1989, Gene, 75: 197-211). To address the reviewer's concern, we exploited a field *A. aegypti* strain from Yunnan province in China (Shi et al., 2016, Acta Parasitol Med Entomol, 23: 175-182), which was collected in 2002. We also used the low-passage clinical DENV-2 43 strain (AF204178) (Liu et al., 2016, Nat Microbiol, 1: 16087) for our investigation. Consistently, knockdown of *AaGABA_A-R1* impaired the infection of DENV-2 43 strain (Supplementary Figure 16A); however, thoracic inoculation of GABA enhanced its infection in the field *A. aegypti* Yunnan mosquitoes (Supplementary Figure 16B). Compared to that of the sucrose meal, feeding either blood (Supplementary Figure 16C) or glutamic acid (Supplementary Figure 16D) substantially enhanced infection of the DENV-2 43 strain in the field *A. aegypti*, suggesting that GABAergic system may play a general role in the promotion of arbovirus infection through blood meals in *A. aegypti* mosquitoes (Page 12, Line 305-315).

As previously mentioned by the reviewer, we determined the vector competence in the experiments that assessed the effect of oral introduction of glutamic acid in arboviral infections. A mixture, which contained 1% sucrose (50% v/v), supernatant from infected Vero cells (cultured in serum-free medium) (50% v/v), and 100 µg/ml glutamic acid, was used to feed mosquitoes via an *in vitro* blood feeding system. Mosquitoes feeding on this mixture without glutamic acid served as a negative control. The midguts, heads and salivary glands were dissected on 7, 14 and 14 days after oral feeding, respectively. The viral loads were measured by qPCR. The ratios of midgut infection, dissemination and transmission were calculated by the Positive Number / the Total Number in the midguts, heads and salivary glands, respectively. In the mosquitoes fed with DENV-2 (Supplementary Figure 15A, D and G), SINV (Supplementary Figure 15B, E and H) and TAHV (Supplementary Figure 15C, F and I),

the oral introduction of glutamic acid significantly enhanced the ratios of midgut infection and dissemination, indicating that glutamate-mediated GABA generation may promote the vector competence for arbovirus infection (Page 11, Line 296-299; Page 12, Line 317-329).

We recognized the reviewer's concerns regarding the viral detection approach. We therefore validated the results of several key experiments with plaque assays. Silencing the *AaGABA_A-R1* gene reduced the number of infectious Dengue-2 virions in the mosquitoes infected by thoracic inoculation (Figure 2B) (Page 6, Line 131-132). Consistently, oral introduction of glutamic acid enhanced the burden of infectious virions of DENV-2 in *A. aegypti* (Supplementary Figure 14A), of SINV in *A. aegypti* (Supplementary Figure 14B), of JEV in *C. pipiens pallens* (Supplementary Figure 14C) and of TAHV in *C. pipiens pallens* (Supplementary Figure 14D), as measured by plaque assays. These results suggested that the qPCR measurement may well reflect the burden of arboviral infectious virions in the mosquitoes (Liu et al., 2016, Nat Microbiol, 1: 16087. Liu et al., 2017, Nature, 545, 482-486) (Page 11, Line 294-295; Page 12, Line 318-323).

#2F. *The abstract, introduction and discussion sections include adaptive evolutionary interpretations that do not make sense. For example, “mechanism employed by mosquitoes to efficiently acquire viruses” (lines 38-39), “mosquitoes have developed some mechanisms that are common to all viruses to facilitate viral infections” (lines 55-56), “a co-evolutionary relationship between mosquito-borne viruses and their vectors, allowing mosquitoes to robustly carry and transmit the viruses” (lines 302-303), give the impression that mosquitoes adapted to become vectors of viruses during evolution. It is unclear (and even counter-intuitive) why mosquitoes would have been selected to become susceptible to viruses, or even why they would have co-evolved with arboviruses.*

#2F Answer: We agreed with the concerns of the reviewer. We have revised these statements in the manuscript (Page 2, Line 37-41; Page 3, Line 55-58; Page 13, Line 355-357).

#2G. *Several figure panels are largely redundant: Fig. 5A and Fig. 5B, Fig. 5C and Fig. 5D, Fig. 6A and Fig. 6B, Fig. 6D and Fig. 6E, Fig. 6F and Fig. 6G, Fig. 6H and Fig. 6I.*

#2G Answer: Thank you for the suggestion. We have revised these figures in the manuscript (Figure 5A, Figure 5B, Figure 5C, Figure 6A, Figure 6C, Figure 6D, Figure 6E, Figure 6F).

#2H. *The information presented in Fig. 1A is unclear. What does this panel show?*

#2H Answer: The data in Figure 1A provides a heatmap of gene regulation, enabling the comprehensive review of gene regulation correlations of different arbovirus infections in *A. aegypti* (Colpitts et al., 2011, PLoS Pathog, 7: e1002189).

#2I. *Survival data should be provided for the GABA injection experiments.*

#2I Answer: We have provided the survival rate data in the response (Figure R2).

#2J. Line 45: “a habit of mosquitoes” should be rephrased as “a behavioral trait of mosquitoes”.

#2J Answer: We have made this change based on the reviewer's suggestion (Page 3, Line 46-47).

#2K. Line 56: “common to all viruses” should be rephrased as “common to all mosquito-borne human viruses”.

#2K Answer: We have made this change based on the reviewer's suggestion (Page 3, Line 56-58).

#2L. Line 92, line 291: According to the American CDC, there are only about a hundred of known arboviruses that are pathogenic to humans.

#2L Answer: We have made this change based on the reviewer's suggestion (Page 5, Line 96).

#2M. Line 95: “dengue” and “encephalitis” should not be capitalized.

#2M Answer: We have made this change based on the reviewer's suggestion (Page 5, Line 99-100).

#2N. Line 102: The number of genes, not the genes.

#2N Answer: We have made this change based on the reviewer's suggestion (Page 5, Line 107-108).

#2O. Line 151: “We then infected those surviving mosquitoes with DENV-2”. This statement is contradictory to what is specified in the legend of Fig. 2CD, which indicates that virus and insecticide were simultaneously inoculated. Please clarify.

#2O Answer: In the Figure 2D and 2E experiments, we co-microinjected a mixture of the insecticides and DENV-2 into the mosquitoes. We have clarified this method in the revised manuscript (Page 7, Line 162-164).

#2P. Line 162: *What was the mode of ZIKV infection? Please specify.*

#2P Answer: In Supplementary Figure 7A and 7B, the mosquitoes were infected by thoracic microinjection. In Supplementary Figure 7C, the mosquitoes were infected by blood feeding with Zika virus. We have clarified these methods in the supplementary figure legends.

#2Q. Line 196: *Why were two different dsRNAs used? Please justify.*

#2Q Answer: dsRNA-mediated gene silencing is a common approach for genetic manipulation in mosquitoes (Balakrishna et al., 2017, *Insect Mol Biol*, 26: 127-139). However, dsRNA inoculation may show off-target effects and non-specifically regulate some genes that are unrelated to the intended target (Moffat et al., 2007, *Trends Pharmacol Sci*, 28: 149-151). We therefore exploited two independent dsRNAs for *AaGABA_A-R1* knockdown, which is a standard requirement for RNAi experiments.

#2R. Lines 225-226: *“the blood meal is an approach by which mosquitoes acquire arboviruses from infected hosts”. Actually, this is the primary mode of mosquito infection by arboviruses.*

#2R Answer: Thank you for the note. We have revised this sentence to "the blood meals are the primary means by which mosquitoes acquire arboviruses from infected hosts" (Page 10, Line 250-251).

#2S. Line 251: *Please specify the infectious dose of virus used to orally infect mosquitoes in Fig. 6, to improve accuracy and allow reproducibility.*

#2S Answer: Thank you for the suggestion. We have added the viral doses in the methods and figure legends (Page 18, Line 490-492; Page 32, Line 853; Page 32, Line 866-868).

#2T. Line 374-377: *It is unclear whether or not 300 nL of liquid was inoculated into the mosquitoes, or if “1 ug per 300 nL” is a concentration. Please correct “injection into virus” on line 377.*

#2T Answer: We thoracically microinjected 1 µg/300 nl dsRNA into each mosquito. We also clarified the viral does used for thoracic microinjection (Page 17, Line 475-477).

#2U. Line 391: *“killed” should be replaced by “sacrificed”.*

#2U Answer: We have made this change based on the reviewer's suggestion (Page 18, Line 494).

#2V. Line 440: *“Flies” must be a typo.*

#2V Answer: We apologized for this mistake. We have made this correction in the revised manuscript (Page 20, Line 553).

Responses to Referee #3:

#3A. *Title and related claims- 'the effects of blood meal acquisition on infection of mosquitoes' are not actually comprehensively studied. To do this, the investigators would need to assure that viral doses to gut epithelial cells acquired through sugar feeding and blood meal feeding were similar, determine the full extent of genes up or downregulated by blood meal digestion and more comprehensively establish the role of each potentially altered gene/pathway on vector competence. Instead, a target was identified through well-supported hypotheses and inoculation studies and the effect of altering that pathway exclusively was studied with blood meal feeding experiments. The title and related language should be changed. Perhaps, 'Blood meal acquisition enhances arbovirus replication in mosquitoes through activation of the GABAergic system' ?*

#3A Answer: We agree the reviewer's suggestion. We have changed the title to "Blood meal acquisition enhances arbovirus replication in mosquitoes through activation of the GABAergic system" (Page 1, Line 1-2).

#3B. *The findings are presented as a discovery of the primary mechanism resulting in permissiveness of mosquitoes to arboviruses. There is no experimental result demonstrating that the absence of GABA completely ablates midgut infection, viral replication or dissemination/transmission (not studied). This result makes sense since Imd, although it clearly has a secondary role, is not the primary immune pathway for defense against RNA viruses. Since GABA has no effect on expression of any molecule which has been implicated in the RNAi pathway, it is not independently capable of facilitating permissiveness to RNA virus infection. Published evidence suggests that this is accomplished by specific viral sequences/proteins evading, suppressing or sequestering the RNAi response, in addition to other mechanisms. There is no mention of the fact that the Imd pathway is primarily used as a defense against Gram negative bacteria. There are multiple statements that seem to suggest that the mosquitoes evolved this system to allow the pathogen to efficiently infect (see line 278). This does not make evolutionary sense. First, even during the most severe outbreaks arboviruses are not prevalent enough to influence mosquito evolution, and second, there is no documented fitness advantage to arbovirus infection in mosquitoes that would drive this. It makes more sense that the pathogen is simply exploiting a system that has evolved for other purposes (perhaps to protect against costly immune activation in the presence of bacteria laden blood). The discussion should be reworked so that the findings are not overstated and the implications fully consider what is known about these systems.*

#3B Answer: Thank you for the reviewer's suggestions and comments. We agree that the primary role of the Imd pathway is in the immune response against gram-negative bacteria rather than for defense against RNA viruses. Therefore, we have added a general description of antiviral mechanisms in mosquitoes, and have subsequently emphasized the primary role of the Imd pathway in anti-bacterial activity (Page 14, Line 370-373; Line 375-380). We also agree with the reviewer's opinion that viruses exploit mosquito factors to facilitate their infection, rather than the idea that

mosquitoes evolved mechanisms to allow the viruses to efficiently infect them. We have changed this statement in the revised manuscript (Page 13, Line 340-342).

#3C. *Doses- Unless I am missing it, there do not seem to be any reports of the doses used to infected mosquitoes in feeding experiments. This is critical information for interpretation.*

#3C Answer: We have added the doses information to the methods and figure legends (Page 18, Line 490-492; Page 29, Line 762; Page 31, Line 832-833; Page 32, Line 853; Page 32, Line 866-868).

#3D. *No assessment of competence- Only fold changes in viral load and proportion infected are compared so the impact on transmission is not clear.*

#3D Answer: To address the reviewer's concern, we determined vector competence in the experiments that assessed the effect of the oral introduction of glutamic acid in arboviral infections. A mixture, which contained 1% sucrose (50% v/v), supernatant from infected Vero cells (cultured in serum-free medium) (50% v/v), and 100 µg/ml glutamic acid, was used to feed mosquitoes via an *in vitro* blood feeding system. Mosquitoes feeding on this mixture without glutamic acid served as negative controls. The midguts, heads and salivary glands were dissected on 7, 14 and 14 days after oral feeding, respectively. The viral loads were measured by qPCR. The ratios of midgut infection, dissemination and transmission were calculated by the Positive Number / the Total Number in the midguts, heads and salivary glands, respectively. In the mosquitoes fed with DENV-2 (Supplementary Figure 15A, D and G), SINV (Supplementary Figure 15B, E and H) and TAHV (Supplementary Figure 15C, F and I), the oral introduction of glutamic acid significantly enhanced the ratios of midgut infection and dissemination, indicating that glutamine-mediated GABA generation may promote the vector competence for arbovirus infection (Page 11, Line 296-299; Page 12, Line 317-329).

#3E. *line 44- In the intro and discussion blood feeding is referred to as a 'common property' of hematophagous insects. That seems to imply that it just happens frequently rather than it being the defining characteristic of these insects.*

#3E Answer: We recognize the reviewer's concern. We have revised these statements to "Feeding on blood is a behavioral trait of mosquitoes that allows them to obtain the necessary nutrients for reproduction" (Page 3, Line 46-48); "Blood meals are the primary means by which mosquitoes acquire arboviruses from infected hosts" (Page 10, Line 250-251); and "Blood acquisition by feeding on a host is a defining characteristic of hematophagous arthropods" (Page 13, Line 333-334).

#3F. *Line 52- Add the word 'Most'. There are pathogenic viruses from other families transmitted by mosquitoes.*

#3F Answer: We have made this change in the revised manuscript (Page 3, Line 54).

#3G. *Line 54- 'Mosquitoes are very permissive' seems to ignore the fact that many are highly refractory and that specific species are responsible for transmission of specific viruses.*

#3G Answer: We have modified this statement in the revised manuscript (Page 3, Line 56-58).

#3H. *Line 84-'facilitating effective systemic viral infection' is an overstatement. Infection is still established in many mosquitoes and, although it would significantly strengthen the significance, dissemination and transmission were not assessed.*

#3H Answer: We recognize the reviewer's concern and have revised the words to "facilitating effective viral infection" (Page 4, Line 85-87). To further address the role of the GABAergic system in the systemic infection, we determined the infectious ratios of midgut infection, dissemination and transmission of arbovirus infection in mosquitoes. In the mosquitoes fed with DENV-2 (Supplementary Figure 15A, D and G), SINV (Supplementary Figure 15B, E and H) and TAHV (Supplementary Figure 15C, F and I), the oral introduction of glutamic acid significantly enhanced the ratios of midgut infection, dissemination and transmission, indicating that glutamate-mediated GABA generation may promote the arbovirus systemic infection and dissemination of arboviruses in mosquitoes. Please refer to the #3D answer.

#3I. *Line 65-6- Dengue and Encephalitis should not be capitalized.*

#3I Answer: We have made this correction.

#3J. *Line 95-100- It is worth noting that only two of the viruses used are transmitted by Ae. aegypti in nature, while 3 are vectored by Culex mosquitoes and one (Batai) by Anopholes. These are really important differences in the interpretation of results. For instance, in figure 1, gene regulation in Ae. aegypti is being compared among viruses which have completely different competence levels in this species. The highest number of genes for which varied expression was measured are with the combination that is most frequent in nature (DENV-aegypti). Somewhat related to this, were virus levels quantified on days 1 and 6? It would be useful to know how viral titer/replication relates to level/spectrum of expression.*

#3J Answer: Based on the reviewer's suggestion, we determined the replication rates of the Dengue (DENV), Japanese Encephalitis (JEV), Semliki Forest (SFV), Sindbis (SINV), Tahyna (TAHV) and Batai (BATV) viruses in *A. aegypti* mosquitoes infected via intrathoracic microinjection. We used 100 MID₅₀ (50% Mosquito Infective Dose) of each virus to infect female mosquitoes. The virus burdens in mosquitoes were quantified by qPCR over a time course post-infection. For the infections of DENV, JEV, TAHV and BATV infections, the viral loads were low on 1 days post-infection (dpi) and subsequently increased from 3 to 9 dpi (Supplementary Figure 1). However, both of the alphaviruses, SINV and SFV, replicated robustly from 1 to 3 dpi, and their viral burdens were then saturated after 3 dpi (Supplementary Figure 1), suggesting that alphaviruses may more aggressively infect and disseminate in *A. aegypti* mosquitoes (Richard et al., 2016, PLoS Negl Trop Dis, 10: e0004694). Based on the infection

rates of the different viruses in the mosquitoes, we analyzed gene regulation on days 1 and 6, which represented early and late infection in the mosquitoes, respectively.

We have noted that *Culex* mosquitoes are the native vectors for JEV, SINV and TAHV used in this study (Kim et al., 2011, Journal of Medical Entomology, 48: 1250-1256. Turell, 2012, J AM Mosq Control Assoc, 28: 123-126. Zhi et al., 2009, Emerg Infect Dis, 10: 130). BATV is carried and transmitted by *Anopheles spp.* (Huhtamo et al., 2013, J Gen Virol, 94: 1242-1248) and *Culex* mosquitoes in nature (Liu et al., 2014, Virol J, 11: 1-8). To investigate the correlation of viral infection-mediated gene regulation between *C. pipiens pallens* and *A. aegypti*, we therefore determined the regulation of the *GABA_A-R1* gene in *C. pipiens pallens* (*CpGABA_A-R1*) infected by JEV, SINV, TAHV and BATV individually. The *GABA_A-R1* gene showed the similar regulation patterns in the infected *C. pipiens pallens* and *A. aegypti*, suggesting that *A. aegypti* may be an appropriate mosquito model to assess gene regulation with arbovirus infections (Figure R3).

#3K. Lines 119-21/suppl fig. 1- There is no further discussion/study of the 2 genes identified as potentially important in suppressing arbovirus infection (i.e. when silenced DENV replication increases). Would seem to be warranted.

#3K Answer: AAEL018153 encodes a hypothetical protein. The biological function of AAEL018153 is not currently understood. AAEL000405 encodes a membrane-anchored cell surface protein named *odd Oz*, which is essential for normal retinal and nervous system development (Kinel-Tahan et al, 2007, Dev Dyn, 239: 2541-2554). We have added these descriptions to the revised manuscript (Page 6, Line 127-129).

#3L. Figure 2 A/B- It is convincing that the titer drop equates to a drop in GABA and this is a sound result. Again, though, authors need to be careful with the biological significance when interpreting. While it is not totally clear what the level of detection is here and how fluorescence equates to viral titer, a 2-4 fold drop is quite small in systems in which titer are normally evaluated on a log₁₀ scale. Similar differences were measured when silencing other genes.

#3L Answer: We recognize the reviewer's concern. We repeated the experiment in Figure 2A using plaque assays. Compared to the control group, the DENV-2 load was reduced by 5-fold in the AaGABA_A-R1-silenced mosquitoes (Figure 2B), suggesting that the qPCR measurements correlate to the number of infectious viruses measured by plaque assays. We added these results to the revised manuscript (Page 6, Line 131-132).

#3M. Lines 230-1- Viruses do not simultaneously infect the gut and spread into the hemocoel as stated. Dissemination is delayed until significant replication in the gut is

achieved and often not attained at all due to the midgut escape barrier. Please restate.

#3M Answer: We agree with the reviewer's suggestion and have restated this in the revised manuscript (Page 10, Line 254-256).

#3N. *Figure 5- Mosquitoes are very unlikely to become fully engorged with a sugar meal as they do with a blood meal. For this reason, it is almost certain that in the sugar meal experiments the viral dose was lower. This could wholly explain the differences that were measured. This needs to be addressed, possibly by quantifying virus at time 0 (following feeding) and possibly adjusting concentrations so that mosquitoes receive similar doses.*

#3N Answer: Thank you for the reviewer's question. In our sucrose feeding system, the 5 to 7-day-old female mosquitoes were fed by 1% sucrose, via a feeder of the Hemotek feeding system. Usually, approximately 90% of the mosquitoes engorged the sugar meal, with more than 90% of them fully engorged. We selected the mosquitoes with full engorgement for the further investigation.

To address the reviewer's question, we compared the size and weight of fully engorged mosquitoes immediately collected after either a sucrose meal or blood feeding. The mosquitoes engorged with sucrose and with a blood meal were similarly sized (Figure R4). Moreover, there were no significant differences in the weights of the mosquitoes fed sucrose or a blood meal (Figure R5). We therefore speculate that the mosquitoes fed a sucrose meal may engorge a similar amount of materials as the mosquitoes fed by blood.

Figure R4. Size measurement of fully engorged mosquitoes immediately collected after either sucrose meal or blood feeding. The 5-7 days old female mosquitoes were fed by either 1% sucrose or whole blood, via a feeder of the Hemotek feeding system. The fully engorged mosquitoes were immediately collected for comparison.

Figure R5. Comparison of the weight of fed mosquitoes by either sucrose meal or blood feeding. The 5-7 days old female mosquitoes were fed by either 1% sucrose or whole blood, via a feeder of the Hemotek feeding system. The fully engorged mosquitoes were immediately collected for comparison.

#30. *Lines 250-2- Sentences beginning on line 250 and 251 seem to contradict. All suppl. Fig 8 shows is that glutamic acid doesn't affect the ability of DENV to form plaques on Vero cell culture. This may or may not tell you anything about infectiousness in vivo in mosquito gut epithelial cells.*

#30 Answer: We recognize the reviewer's concern and revised this in the manuscript (Page 11, Line 293-294). In the experiment of Supplementary Figure 13, we incubated DENV-2 with a serial concentration of glutamic acid for 30 minutes. DENV-2 infectivity after *in vitro* glutamic acid treatment was measured by a plaque assay in Vero cells.

This experiment is intended to address whether glutamic acid directly influences DENV virions. The effect of glutamic acid on the infection of mosquito guts (the *in vivo* assay) was determined and is presented in Supplementary Figure 14A. These data, together with others data, suggest that glutamic acid enhances the DENV-2 infection of mosquitoes by modulating the mosquito immune system, not by directly influencing the infectiousness of DENV virions.

#3P. *Line 284-Remove the word 'dissemination'. This is not evaluated in the study.*

#3P Answer: The word "dissemination" has been removed (Page 13, Line 348-350)

#3Q. *Line 294-Again, mosquitoes don't 'exploit' these receptors, pathogens exploit them. This seems to be consistent with many statements suggesting there is an evolutionary advantage for the mosquito to be permissive to arboviruses*

#3Q Answer: We recognize the reviewer's concern. We have revised this statement in the revised manuscript (Page 3, Line 56-58; Page 13, Line 340-342).

Reviewers' comments:

Reviewer #1 (Remarks to the Author):

The authors have done a thorough job adding information to this manuscript and answered my points (and going through the responses, also the relevant comments made by other reviewers). It may be useful to add the figures that were included in the Response to reviewers documents. I would suggest to add the figures included in the response document, although I appreciate that this is already a very detailed paper. The authors should ensure virus names are correct (dengue not Dengue, encephalitis, o'nyong-nyong not O'nyong).

Reviewer #2 (Remarks to the Author):

The new data produced by the authors has alleviated most of the reviewers' concerns, however there is still at least one important point to be addressed.

Whereas Fig. 6 shows that GABA enhances virus infection through sugar feeding, the data presented in Fig. 5 could be misleading because the 1% sucrose controls are likely invalid. Indeed, during sugar feeding most mosquitoes store sucrose in a sack-like crop distinct from their gut. Therefore, what the authors may have compared in these experiments is the mosquito oral susceptibility to virus infection through the crop versus the midgut. The observed difference in oral susceptibility may have nothing to do with the GABAergic system but rather reflect tissue-specific differences in susceptibility. This might explain why silencing of AaGAD1 only partially offsets the enhancing effect of blood feeding on infection (Fig. S11). The authors should check whether the virus-spiked sucrose meal is ingested into the crop and/or the midgut and clearly discuss this issue in their paper.

Fig. S11 and Fig. S16C: why do so many mosquitoes have the same DENV/Actin mRNA ratio in the Blood and GFP+Blood groups, respectively?

Throughout the manuscript there are several typos that need to be corrected.

Reviewer #3 (Remarks to the Author):

The authors have carefully responded to individual comments or critiques. In all instances, either interpretations or language was adequately edited to match the data or additional experiments were completed to more thoroughly support the conclusions. The findings are novel and of broad interest and, as presented, are now adequately supported by the data.

Response to Referee #1:

#1A. It may be useful to add the figures that were included in the Response to reviewers documents. I would suggest to add the figures included in the response document, although I appreciate that this is already a very detailed paper.

#1A Answer: Thanks for the reviewer's suggestion. We have added the Figures in the Response document to the manuscript (Supplementary Figure 4, Supplementary Figure 7A, Supplementary Figure 9, Supplementary Figure 13A-B, Supplementary Figure 14)

#1B. The authors should ensure virus names are correct (dengue not Dengue, encephalitis, o'nyong-nyong not O'nyong).

#1B Answer: We have made the corrections in the revised manuscript (Page 5, Line 100; Page 15, Line 393)

Response to Referee #2:

#2A. Whereas Fig. 6 shows that GABA enhances virus infection through sugar feeding, the data presented in Fig. 5 could be misleading because the 1% sucrose controls are likely invalid. Indeed, during sugar feeding most mosquitoes store sucrose in a sack-like crop distinct from their gut. Therefore, what the authors may have compared in these experiments is the mosquito oral susceptibility to virus infection through the crop versus the midgut. The observed difference in oral susceptibility may have nothing to do with the GABAergic system but rather reflect tissue-specific differences in susceptibility. This might explain why silencing of AaGAD1 only partially offsets the enhancing effect of blood feeding on infection (Fig. S11). The authors should check whether the virus-spiked sucrose meal is ingested into the crop and/or the midgut and clearly discuss this issue in their paper.

#2A Answer: We recognize the reviewer's concern. In the Figure 5A-C, the mosquitoes were fed by the mixture containing either 1% sucrose or human blood (50% v/v) with the supernatant from virus-infected Vero cells (cultured in serum-free medium) (50% v/v). The reviewer questioned that the sucrose liquid might be acquired into a sack-like crop rather than the midgut by a sucrose meal. Indeed, the previous studies suggested that acquisition and digestion of sugar in insects may take place in either the midgut or the crop (Dong et al., 2009, PLoS Pathog, 5: e1000423; Gendrin et al., 2014, Nature Communications, 6: 5921; Ondiaka et al., 2015, Parasit Vectors, 8: 35). However, the digestion location and rate is influenced by the meal size consumed, sugar concentration and other factors (Handel, 1965, J. Physiol, 181: 478-486). In this study, we fed the mosquitoes with a mixture of 1% sucrose and cell culture medium at a 1:1 ratio, rather than a pure sugar solution. To address the reviewer's concern, we therefore investigated which tissue is exactly used for stocking the virus-spiked sucrose liquid after a meal.

In the experiment, the *Aedes aegypti* mosquitoes were fed by 1% sucrose (50% v/v), the supernatant from DENV-2-infected Vero cell culture (50% v/v), with 0.1% Ponceau S (the final concentration). Both the midgut and the crop were dissected at 0, 2, 4, 8 hours after the sucrose meal. From the results, we can clearly observe that the liquid was immediately acquired into the midguts rather than the crop tissues after a meal (Figure R1). Though the sucrose liquid was quickly consumed, the red dye still remained in the midgut at 4 and 8 hours post meal, indicating the materials introduced by the sucrose meal will be stocked in the midgut. Given that the blood is directly drawn into the mosquito midgut through blood feeding, we concluded that the viral infectivity is comparable between sucrose- and blood-feeding mosquitoes. We have added the result in the revised manuscript (Page 10, Line 269; Supplementary Figure 14).

Indeed, the sucrose meal is a common approach to orally introduce external materials into the mosquito midgut. The commensal microbes were removed from the mosquito gut by antibiotic feeding with a sucrose meal (Xi et al., 2008, PloS Pathog, 4: e1000098; Meister et al., 2009, PloS Pathog, 5: e1000542; Dong et al., 2009, PloS Pathog, 5: e1000423; Gendrin et al., 2014, Nature Communications, 6: 5921). The previous literature also suggested that the digestion of sugar meal in mosquitoes took place in the midgut when the gut sugar content was measured by the cold anthrone test (Hien et al., 2016, PloS Pathog, 12: e1005773), suggesting the sugar meal is a feasible approach for oral introduction of external material into the mosquito midgut.

#2B. Fig. S11 and Fig. S16C: why do so many mosquitoes have the same DENV/Actin mRNA ratio in the Blood and GFP+Blood groups, respectively?

#2B Answer: To address the reviewer's concern, we presented the source data of Supplementary Figure 15 (the original Fig. S11) and Supplementary Figure 20C (the original Fig. S16C). We found that the scale format of the Y-axis concentrated the dots in the original Figures. Therefore, we re-set the Y-axis scale to scatter the dots to avoid potential misunderstanding by readers (Figure R2) (Supplementary Figure 15, Supplementary Figure 20C).

Figure R1. Tracking the acquisition of sucrose liquid in the mosquito midgut and crop. The *A. aegypti* mosquitoes were fed by 1% sucrose (50% v/v), the supernatant from DENV-2-infected Vero cells (50% v/v), with 0.1% Ponceau S (the final concentration). Both the midgut and the crop were dissected for investigation at 0, 2, 4, 8 hours after the sucrose meal.

Figure R2. The source data of Supplementary Figure 15 and Supplementary 20C

#2C. Throughout the manuscript there are several typos that need to be corrected.

#2C Answer: We have checked the manuscript and corrected some typos in the text (Page 4, Line 84; Page 5, Line 100; Page 6, Line 145; Page 11, Line 276; Page 15, Line 393).

Thank you for the helpful comments to improve this manuscript.

REVIEWERS' COMMENTS:

Reviewer #2 (Remarks to the Author):

The authors satisfactorily addressed my last concerns. Congratulations on a nice study.